# Focal optogenetic suppression in macaque area MT biases direction discrimination and decision confidence, but only transiently

Christopher R Fetsch[1,2]*, Naomi N Odean[3,4,5], Danique Jeurissen[3,4,5], Yasmine El-Shamayleh[6], Gregory D Horwitz[6], Michael N Shadlen[3,4,5]*

[1]Zanvyl Krieger Mind/Brain Institute, Johns Hopkins University, Baltimore, United States; [2]Solomon H. Snyder Department of Neuroscience, Johns Hopkins University, Baltimore, United States; [3]Kavli Institute, Columbia University, New York, United States; [4]Howard Hughes Medical Institute, Columbia University, New York, United States; [5]Department of Neuroscience, Zuckerman Mind Brain Behavior Institute, Columbia University, New York, United States; [6]Department of Physiology & Biophysics, Washington National Primate Research Center, University of Washington, Washington, United States

**Abstract** Insights from causal manipulations of brain activity depend on targeting the spatial and temporal scales most relevant for behavior. Using a sensitive perceptual decision task in monkeys, we examined the effects of rapid, reversible inactivation on a spatial scale previously achieved only with electrical microstimulation. Inactivating groups of similarly tuned neurons in area MT produced systematic effects on choice and confidence. Behavioral effects were attenuated over the course of each session, suggesting compensatory adjustments in the downstream readout of MT over tens of minutes. Compensation also occurred on a sub-second time scale: behavior was largely unaffected when the visual stimulus (and concurrent suppression) lasted longer than 350 ms. These trends were similar for choice and confidence, consistent with the idea of a common mechanism underlying both measures. The findings demonstrate the utility of hyperpolarizing opsins for linking neural population activity at fine spatial and temporal scales to cognitive functions in primates.

DOI: https://doi.org/10.7554/eLife.36523.001

*For correspondence:
cfetsch@jhu.edu (CRF);
shadlen@columbia.edu (MNS)

**Competing interests:** The authors declare that no competing interests exist.

## Introduction

To understand how neural activity gives rise to behavior, a powerful approach is to manipulate the activity of groups of neurons defined by particular functional or anatomical properties in the context of a suitable behavioral task. This strategy has grown in popularity over recent years with the advent of sophisticated tools for manipulating neural circuit function. However, recent perspectives have cautioned that so-called causal evidence is not always as decisive as it may appear (*Jazayeri and Afraz, 2017*), and by itself does not generate the level of understanding we wish to attain (*Krakauer et al., 2017*). These and other arguments serve to renew a longstanding dictum in systems neuroscience, namely the primary importance of developing a rigorous theoretical or conceptual framework for understanding the behavior of interest. Only then can clear hypotheses be articulated about the causal role of neural populations or circuits in generating the behavior.

The study of perceptual decision making has achieved a degree of progress toward this goal, in part by leveraging detailed knowledge of the neural representation of sensory evidence supporting

the decision (reviewed by *Cohen and Newsome, 2004*; *Romo and Salinas, 2001*; *Shadlen and Kiani, 2013*). One well-studied behavioral task requires judgment of the net direction of motion in a noisy visual display designed to promote the integration of motion information across the display and over time. Neurons in motion-sensitive cortical areas, especially the middle temporal (MT) and medial superior temporal (MST) areas, are well suited to provide the evidence for the task, and theoretical considerations (*Gold and Shadlen, 2001*; *Shadlen et al., 1996*) point toward a simple and plausible computation: the subtraction of spike counts (or rates) between pools of neurons favoring each of the direction alternatives. This difference furnishes a quantity proportional to the log likelihood ratio favoring a given alternative, a statistical measure of the weight of evidence that can be accumulated over time to support optimal statistical decision making (*Gold and Shadlen, 2002*; *Wald, 1947*)

Building on the work of Newsome and colleagues (*Celebrini and Newsome, 1995*; *Salzman et al., 1992*), *Ditterich et al. (2003)* took advantage of the columnar organization of MT (*Albright et al., 1984*) to demonstrate, beyond simply a causal role for these neurons in the task, the specific differencing-and-integration mechanism that was previously hypothesized on theoretical grounds. Using electrical microstimulation (µStim) to activate neurons that were largely within a single direction column, they confirmed earlier reports showing that monkeys' choices were biased toward the preferred direction of the activated neurons. More importantly, µStim increased the speed of decisions in favor of the preferred direction while slowing decisions made in favor of the opposite direction. The results supported a model in which the decision is formed by temporal integration of momentary evidence, defined as the difference in activity of pools of MT neurons tuned for the preferred and anti-preferred motion direction (*Ditterich et al., 2003*). More recently, we showed that decision confidence is affected by µStim in a way that is quantitatively commensurate with the effect on choices and well explained by a bounded evidence accumulation model (*Fetsch et al., 2014a*). Taken together, these findings suggest that a common process of evidence accumulation underlies all three behavioral measures of decision making in this task: choice, reaction time, and confidence.

Causal activation of neurons informs our understanding of the sufficiency, but not the necessity, of neural activity for driving behavior. Thus, reversible inactivation offers an important complement to stimulation, but conventional methods in primate neuroscience (e.g., muscimol or cooling) are lacking in both spatial and temporal resolution. The temporal component is crucial and often neglected. Indeed, the insights gained from the µStim studies described above depended on targeting the perturbation not only to the appropriate neurons but also the appropriate time frame within a trial (*Seidemann et al., 1998*). µStim itself has other drawbacks, including the possibility of antidromic activation, the generation of unwanted temporal and spatial patterns of *in*activation (*Butovas and Schwarz, 2003*; *Logothetis et al., 2010*; *Seidemann et al., 2002*), and the difficulty of quantifying concurrent changes in neural activity due to voltage artifacts (but see *Gnadt et al., 2003*).

To address these limitations, we used an optogenetic approach to suppress MT activity during a motion discrimination task with post-decision wagering (PDW; *Figure 1A*). While some primate optogenetics studies set out to illuminate the largest possible volume of tissue (*Acker et al., 2016*; *Gerits et al., 2012*), we used very low light power levels to target relatively small clusters of excitatory neurons with similar selectivity for direction of motion. The rationale for this approach is that, in MT, such clusters constitute a critical functional unit for the conversion of a sensory representation into evidence for the decision. We found that optogenetic suppression was capable of inducing a choice bias against the neurons' preferred direction, and a qualitatively similar change in the pattern of post-decision wagering, broadly consistent with a common mechanism underlying choice and confidence. We also found intriguing evidence for compensatory changes in the readout of MT activity, on both the sub-second and tens-of-minutes time scales. The results highlight the importance of spatial and temporal specificity in causal interventions, and they suggest a possible explanation for weak behavioral effects of optogenetic manipulations in some previous nonhuman primate studies.

## Results

We examined the effects of optogenetic inactivation (hereafter, photosuppression) in extrastriate visual cortex (area MT) on perceptual choices and decision confidence. Two monkeys were trained

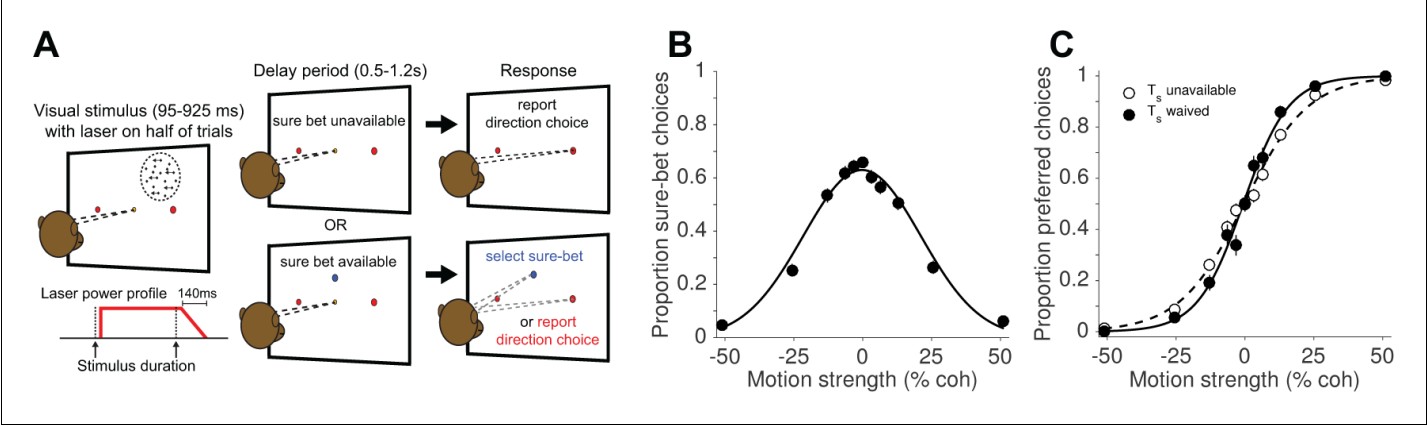

**Figure 1.** Direction discrimination task with post-decision wagering (PDW). (**A**) The monkey fixates on a central fixation point to initiate the trial. Two red choice targets are presented, followed by a random-dot motion (RDM) stimulus in the receptive field of the recorded neurons (left panel), shown for an experimenter-controlled variable duration (*Figure 1—figure supplement 1*). On a random half of trials, a 'sure-bet' target ($T_s$) is presented (blue spot in the lower panels). After a delay period, the monkey can report his choice by a leftward or rightward saccade to one of the red targets to obtain a large juice reward if he is correct, or, if available, choose the sure-bet target for a small but guaranteed reward. On half of trials, including both $T_s$-present and $T_s$-absent trials, the RDM stimulus was accompanied by red laser illumination (step-rampdown power profile) of a cluster of neurons expressing the light-sensitive chloride pump Jaws. (**B**) Proportion sure-bet choices as a function of motion strength ('confidence function') for all no-laser trials (N = 2 monkeys, 23 sessions, 9912 trials). Error bars in B and C indicate standard errors of the proportions and are often smaller than the data points. (**C**) Proportion of preferred-direction choices as a function of motion strength ('choice function') for all no-laser trials, separated by whether the sure-bet option was unavailable (dashed) or available but waived (solid).

DOI: https://doi.org/10.7554/eLife.36523.002

The following source data and figure supplement are available for figure 1:

**Source data 1.** Data and Matlab code for reproducing all panels and figure supplements for *Figure 1*.
DOI: https://doi.org/10.7554/eLife.36523.004
**Figure supplement 1.** Distribution of viewing duration and its effect on confidence.
DOI: https://doi.org/10.7554/eLife.36523.003

to perform a random-dot motion (RDM) direction discrimination task with post-decision wagering (PDW; *Figure 1A*). The task was to decide whether the net direction of dot motion was to the left or to the right, and to indicate the choice with a saccade to a leftward or rightward target when prompted. In addition to the choice targets, a 'sure-bet' target was presented during the delay period on a random half of trials, allowing the monkey to receive a guaranteed but smaller reward and thereby indicate its lack of confidence in the binary left-right decision. As in previous studies (*Fetsch et al., 2014a*; *Kiani and Shadlen, 2009*; *Zylberberg et al., 2016*), monkeys chose the sure-bet most frequently when the motion was weak (*Figure 1B*) and of short duration (*Figure 1—figure supplement 1B*), and their accuracy was greater when the sure-bet was available but waived, versus when it was unavailable (*Figure 1C*; $p < 10^{-10}$, logistic regression). These behavioral observations, and quantitative analyses published previously (*Fetsch et al., 2014a*; *Kiani and Shadlen, 2009*), serve to validate the PDW assay as a measure of confidence—that is, a prediction of accuracy based on the state of a decision variable on a given trial, rather than a low-level estimate of trial difficulty or an index of lapses of attention.

## Histological and physiological characterization of Jaws expression

At least 8 weeks before commencing experiments, area MT in one hemisphere of each animal was injected with an AAV vector to drive expression of the red light-sensitive chloride pump Jaws (cruxhalorhodopsin; *Chuong et al., 2014*) under the control of the CaMKIIα promoter. The two animals continue to participate in experiments and are unavailable for histology. Therefore, to test the overall efficacy of the AAV-CaMKIIα-Jaws vector (to our knowledge, not previously used in the macaque), we performed immunohistochemical analysis in a third animal following injections in the lateral intraparietal area (LIP). This analysis revealed robust Jaws-GFP expression in superficial and deep layers (*Figure 2A*), and a tropism for excitatory pyramidal cells (*Figure 2B*), as shown in previous studies

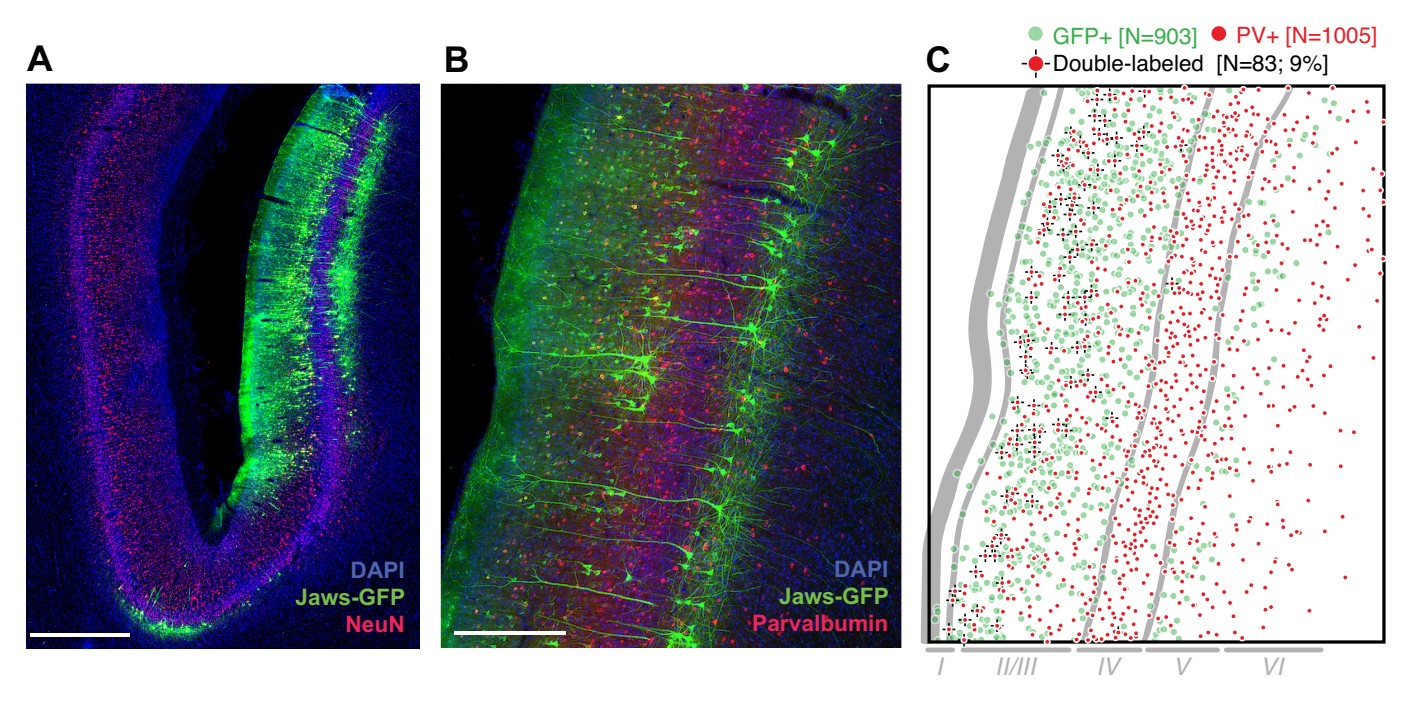

**Figure 2.** Transduction of neurons in macaque lateral intraparietal area (LIP) by AAV-CamKIIα-Jaws-KGC-GFP-ER2. (**A**) Histological section imaged at 10X showing expression of Jaws (green) following a series of viral vector injections along a single injectrode penetration in area LIP (different monkey and brain area than was tested in experiments described below). Expression was strongest in layers II/III and V. Red: NeuN. Blue: DAPI. Scale bar = 1 mm. (**B**) 20X image of a nearby section stained for Jaws-GFP (green) and the inhibitory marker parvalbumin (PV, red). Scale bar = 500 μm. (**C**) Cell counts from the image in B, quantifying the small but nontrivial minority of neurons double-labeled for Jaws-GFP and PV. Approximate layer boundaries are indicated in gray.

DOI: https://doi.org/10.7554/eLife.36523.005

using the CaMKIIα promoter (*Han et al., 2009*; *Nassi et al., 2015*). This tropism was not as strong as in previous work: 9% of Jaws-GFP-positive neurons (83 of 903) were double-labeled for the inhibitory marker parvalbumin (*Figure 2B*), giving an upper bound on selectivity of 91%, compared to >98% in macaque primary visual cortex (2 of 119 cells double-labeled for any of three different inhibitory markers; *Nassi et al., 2015*), and 100% in a study of the frontal eye field (0 of 78 cells double-labeled for GABA; *Han et al., 2009*). However, in addition to being conducted in different cortical areas, these studies used lentiviral vectors rather than AAV vectors. Thus, the effectiveness of promoter-based targeting of excitatory neurons likely depends on the viral vector and/or serotype (*Nathanson et al., 2009*; *Gerits et al., 2015*), and could also vary across brain areas. We emphasize that any conclusions from *Figure 2* in relation to the results presented below should be rendered with caution, considering possible differences in cell-type distributions and laminar organization of MT versus LIP.

In each experimental session, we advanced a custom optrode into area MT and began searching for candidate sites for testing the effects of photosuppression on behavior. If a suitable site was found (see Materials and methods), the monkey commenced the task described above, with low-power red illumination (633 nm, total power = 0.25–2.0 mW, irradiance = 2–16 mW/mm² at a distance of 300 μm) delivered concurrently with visual stimulation (laser delayed by a 20 ms) on a random half of trials. Note that unlike previous studies, a red-shifted opsin was chosen not for its suitability for large-volume tissue illumination (*Acker et al., 2016*) but for its superior photocurrent at very low irradiance (*Chuong et al., 2014*). Optrode design, laser power, and site selection criteria were all intended to limit the light-induced suppression of activity to a relatively small cluster of MT neurons with consistent receptive field and tuning properties (i.e., preferred speed and direction of motion). We estimated tissue irradiance as a function of laser power and distance from the fiber tip

using available online tools (https://web.stanford.edu/group/dlab/cgi-bin/graph/chart.php) as well as Monte Carlo simulations (*Stujenske et al., 2015*) (*Figure 3—figure supplement 1*). These estimates suggest that physiologically effective irradiance levels extended at most 300–500 µm from the fiber tip, which we confirmed qualitatively in pilot experiments using a fiber affixed to a linear electrode array (V-Probe, Plexon, Inc., Dallas, TX). After identifying a candidate site, we titrated the laser power to achieve an amount of suppression just shy of maximum (saturating) levels (*Figure 3—figure supplement 2*).

During the mapping procedure, we occasionally isolated single neurons and used this opportunity to better quantify the physiological effects of the light. Single-unit visual responses to high-coherence RDM stimuli—presented at the preferred direction and speed—were strongly suppressed on average (N = 26; *Figure 3A*), although a handful of neurons showed no effect or an increase in firing rate. Presumably this handful of cells includes those that did not express Jaws, as well as a small minority that were indirectly activated through polysynaptic mechanisms. Among putative Jaws-expressing neurons, the visually evoked response, after subtracting baseline activity, was suppressed by 101% (±13% SEM; N = 20; *Figure 3B*). Without baseline subtraction, the average firing rate during stimulus presentation was reduced by 72% ± 5%. In contrast, MU responses during the discrimination task were reduced by an average of 33% (±0.3% SEM, N = 20058 trials; *Figure 3C*). The fractional degree of suppression (*Equation 4*; Materials and methods), calculated from MU activity, varied substantially both within and across sessions (*Figure 3A*, inset; *Figure 4A*). We take this variability into consideration in the analyses to follow.

Note that the rebound of spiking after abrupt laser offset (*Figure 3B*), a known byproduct of photosuppression, was mitigated by introducing a 140 ms ramp-down of laser power in the majority of behavioral sessions (*Figure 3D*; *Chuong et al., 2014*). We did not detect a difference in behavioral effects when the rebound was suppressed, compared to an earlier subset of sessions without the ramp-down (*Figure 3D*, inset), and therefore pooled all sessions for subsequent analyses.

## Behavioral effects of photosuppression

To introduce the behavioral analyses presented below, we first revisit results from recent µStim experiments using the same behavioral task (*Fetsch et al., 2014a*). Microstimulation in areas MT and MST altered choice and confidence in a manner that largely mimics a change in the motion strength, with a directionality predicted by the preferred direction of the stimulated neurons (*Figure 4—figure supplement 1*). Although there were subtle changes in the slope of the choice function, and the height and width of the confidence function (see Discussion), here for simplicity we focus on the lateral shift of the functions in units of motion strength (% coherence; *Equations 1 and 3*, Materials and methods). The direction and size of these shifts in the previous study were well matched in the aggregate, and highly correlated across sessions, suggesting that MT/MST activity contributes to a common decision variable that governs both choice and confidence (*Fetsch et al., 2014a*; *Kiani and Shadlen, 2009*).

Under this interpretation, suppression of MT activity—provided it is targeted to a population of neurons that share a common tuning preference—should cause shifts in the opposite direction: fewer choices in the preferred direction, and a rightward shift of the bell-shaped confidence function. Moreover, one would expect behavioral effects to depend on the degree of suppression, which as noted above, was highly variable. For all laser trials in each session, we calculated the fractional change in multi-unit firing rate ($\Delta R$; *Equation 4*) relative to the mean of no-laser trials for the corresponding session and trial type (motion direction and coherence; *Figure 4A*). We then estimated the behavioral effect size (*Equations 1 and 3*) as a function of neuronal effect size, using a sliding window of trials pooled across sessions and sorted by $\Delta R$. This analysis (*Figure 4B*) showed that photosuppression caused a modest but significant choice bias in the predicted direction, provided that the degree of suppression was greater than about 45% (*Figure 4C*; shift = −2.1% coh±0.6% SE; p=0.001). The relationship between $\Delta R$ and the effect on confidence was more complex (*Figure 4B*, blue trace), but the largest effects were also observed when suppression was strong (*Figure 4D*; shift = −2.9% ± 1.4%, p=0.03). We therefore limited most of the following analyses to trials showing clear effects on neural activity, using an arbitrary cutoff of 25% ($\Delta R < -0.25$, unless noted), rather than 45%, to increase statistical power.

To address potential contributions of neuronal response variability, independent of variability in the degree of suppression, we performed the following control analysis. Taking only no-laser trials,

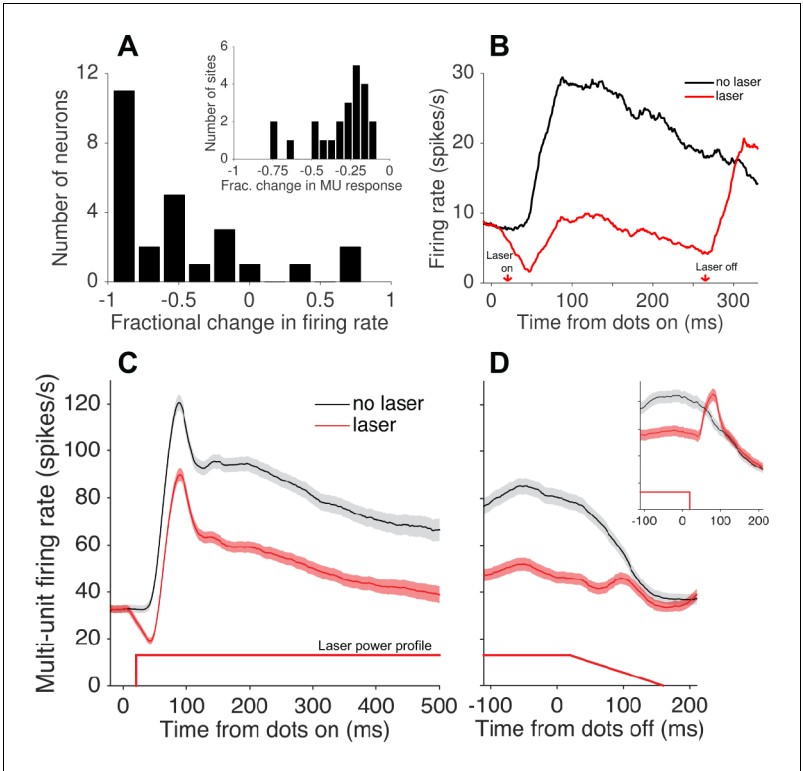

**Figure 3.** Photosupression of neural activity in area MT. (**A**) Fractional change in firing rate of isolated single units (N = 26) in response to a high-coherence RDM stimulus and laser suppression, relative to RDM stimulus alone. Single-unit data were collected during a fixation task. The inset shows the fractional change in multi-unit activity recorded at the 23 sites tested in the discrimination task. (**B**) Average firing rate (peristimulus time histogram) of single units showing significant Jaws-mediated suppression (N = 20). Laser onset occurred 20 ms after stimulus onset, resulting in suppression that preceded the onset of visually-driven activity (i.e., firing rate driven below baseline, then recovered to near-baseline levels during visual stimulation). A post-suppression rebound of activity was observed after turning off the laser (and visual stimulus). All PSTHs depict spike counts in 1 ms bins convolved with a 40 ms causal boxcar filter and converted to spikes/s (**C, D**) Average multi-unit activity for N = 23 sites passing the selection criteria for the behavioral experiment (see Materials and methods), aligned to stimulus (dots) onset (**C**) and offset (**D**). The majority of sessions included a ramp-down of 140 ms in laser power starting 20 ms after stimulus offset, reducing the post-suppression burst seen when no ramp-down was used (inset). Shaded region shows ± SEM. *Figure 3—figure supplement 1* shows an estimated spatial distribution of irradiance based on Monte Carlo simulations of light transmission in brain tissue, and *Figure 3—figure supplement 2* illustrates an example of varying laser power at a fixed distance.
DOI: https://doi.org/10.7554/eLife.36523.006

The following source data and figure supplements are available for figure 3:

**Source data 1.** Data and Matlab code for reproducing all panels and figure supplements for *Figure 3*.
DOI: https://doi.org/10.7554/eLife.36523.009
**Figure supplement 1.** Predicted irradiance based on Monte Carlo simulations of light transmission in brain tissue (validated with previously published in-vivo measurements; *Stujenske et al., 2015*), for three different levels of total laser power: 0.3 mW (**A**), 0.7 mW (**B**), and 1.2 mW (**C**) at a wavelength of 633 nm.
DOI: https://doi.org/10.7554/eLife.36523.007
**Figure supplement 2.** Titration of laser power.
DOI: https://doi.org/10.7554/eLife.36523.008

we randomly assigned 50% of them a fictitious category ("sham laser trials"), sorted these trials as above by their fractional difference (ΔR) relative to the mean for a given session and trial type (*Figure 4—figure supplement 2A*), and performed the same sliding-window analysis of choice bias and shifts of the confidence function. We then repeated the procedure 100 times with different randomized trial assignments and averaged the results. The resulting traces (*Figure 4—figure supplement*

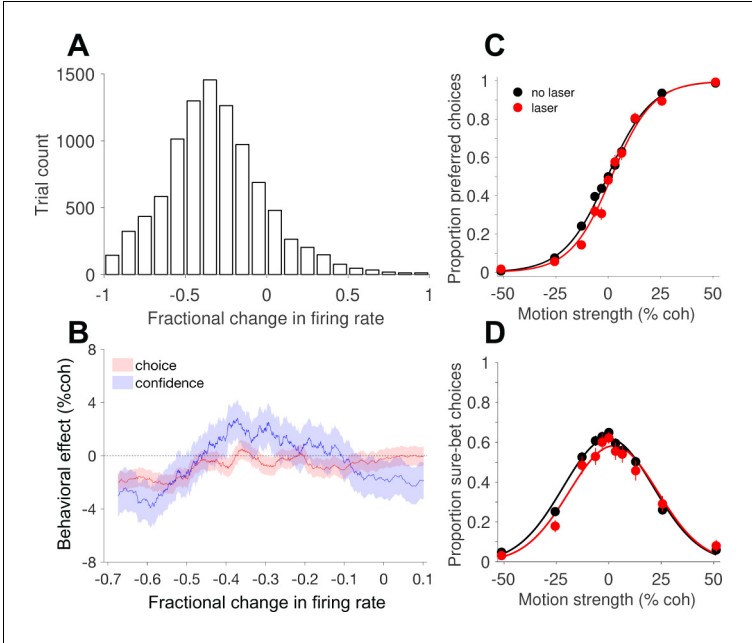

**Figure 4.** Behavioral effects of photosuppression depend on the magnitude of reduction in neural activity. (**A**) Histogram of the fractional change in multi-unit firing rate ($\Delta R$) on laser trials, relative to the mean on no-laser trials for the corresponding session and trial type (**Equation 4**). (**B**) Behavioral effects, expressed as the horizontal shift in the choice function (**Equation 1**, red) and bell-shaped confidence function (**Equation 3**, blue), plotted as a function of $\Delta R$. Each abscissa coordinate represents the mean of a sliding quantile, hence all the data in A are represented even though the abscissa range is different. Shaded error regions indicate ± SEM. **Figure 4—figure supplement 2** shows the results of a corresponding control analysis of no-laser trials. (**C**) Choice functions for the set of laser trials (red) exhibiting strong suppression ($\Delta R < -0.45$, N = 3107 trials) compared to all no-laser trials (black, N = 9249 trials). (**D**) The corresponding confidence functions (proportion sure-bet choices as a function of motion strength) for the set of trials shown in C. For comparison, **Figure 4—figure supplement 1** shows the effects of electrical microstimulation from previous experiments in monkey D, and **Figure 4—figure supplement 3** shows the distribution of effects session by session, including all trials irrespective of $\Delta R$ and other influencing factors (see below).

DOI: https://doi.org/10.7554/eLife.36523.010

The following source data and figure supplements are available for figure 4:

**Source data 1.** Data and Matlab code for reproducing all panels and figure supplements for **Figure 4**.
DOI: https://doi.org/10.7554/eLife.36523.014

**Figure supplement 1.** Effects of electrical microstimulation (µStim) on choice and confidence.
DOI: https://doi.org/10.7554/eLife.36523.011

**Figure supplement 2.** Sham-laser control analysis.
DOI: https://doi.org/10.7554/eLife.36523.012

**Figure supplement 3.** Behavioral effects on individual sessions.
DOI: https://doi.org/10.7554/eLife.36523.013

---

**2B**) showed a weak trend consistent with a covariation between response fluctuations and behavior (i.e., choice probability; **Britten et al., 1996**), but which was several fold smaller than in the original analysis that included actual photosuppression (**Figure 4B**) and not statistically significant (p=0.4, **Equation 5**, $\beta_6$). In direct comparison with the effects shown in **Figure 4C–D**, the behavioral shifts on a selected group of control trials with matching $\Delta R$ (i.e., less than -0.45) were significantly smaller ($p<10^{-5}$ for choice and p = 0.02 for confidence). This result implies that the behavioral effects we observed under strong photosuppression cannot be explained away by normal response variability and its covariation with choice (and confidence).

## Temporal factors influence behavioral effects: compensation on multiple time scales?

The result shown in *Figure 4B* implies that photosuppression was only effective at altering behavior for a subset of trials. Indeed, most individual sessions failed to show statistically reliable effects on behavior when analyzed in their entirety (*Figure 4—figure supplement 3*). However, while collecting the data we noticed a tendency for systematic biases on laser trials to appear early within a session, only to dissipate over the course of tens of minutes (100–500 trials; ~5 s per trial on average, including intertrial intervals). This can be seen in the example session shown in *Figure 5*. Over the first 500 trials, the monkey tended to choose the preferred direction less often on laser trials vs. no-laser trials, effectively shifting the psychometric curve to the right (*Figure 5A*; shift = −4.6%, p=0.14, *Equation 1*). In contrast, there was essentially no effect on choices in the remaining ~1000 trials of the session (*Figure 5B*; shift = −0.7% coh, p=0.75). The pattern was similar for the confidence assay: a rightward shift of the confidence function on early trials (*Figure 5C*; shift = −4.2%, p=0.34) but not later trials (*Figure 5D*, 1.2%, p=0.80). This trend was evident when pooling across all sessions with enough trials to measure it (minimum 800 trials per session, N = 10 sessions): rightward shifts early in the session (choice shift = −3.5% coh±1.1%, p=0.001, *Figure 5A'*; confidence shift = −3.8% ± 1.9% [*Equation 3*], p=0.05, *Figure 5C'*) but not later (choice shift = −0.7%, p=0.26, *Figure 5B'*; confidence shift = 0.1%, p=0.93; *Figure 5D'*). Post-hoc tests using generalized linear models (*Equation 5*, see Materials and methods) showed that the difference between early and late trials was statistically significant for choice (p=0.01) and confidence (p=0.02, Bonferroni corrected).

*Salzman et al. (1992)* reported that the effects of μStim dissipated over the course of the behavioral sessions, which they interpreted as indicating neuronal fatigue, damage, or shifts in the position of the stimulating electrode. In our case, however, the decrease in effectiveness of photosuppression on behavior was not accompanied by an attenuation of its effect on neural responses. In the example session (*Figure 5E* vs. *Figure 5F*), the degree of suppression was slightly greater in early versus late trials (ΔR = −0.80 ± 0.01 and −0.74 ± 0.01, respectively; p<0.05, bootstrap confidence interval, resampling trials with replacement), but this quantitative difference seems unlikely to explain the complete nullification of the behavioral effects (*Figure 5B,D*). Across all 10 sessions in the pooled dataset (*Figure 5E'* vs. *Figure 5F'*) the trend was in the opposite direction, toward greater suppression on late trials (ΔR early = −0.28 ± 0.008, late = −0.40 ± 0.004, p<0.05). Furthermore, we did not observe a qualitative change in the direction selectivity of the affected neurons between early and late trials (*Figure 5G* vs. *Figure 5H*; and *Figure 5G'* vs. *Figure 5H'*). At the conclusion of a subset of sessions, we re-measured direction tuning at multiple depths and found little evidence for optrode drift, tissue damage or response fatigue. Instead, the results suggest a compensatory change in the downstream readout of MT neural signals to reduce or nullify the effects of photosuppression on a time scale of 10–40 minutes.

To examine this putative compensation at a finer grain, we plotted the choice and confidence effects calculated in a sliding window of trials sorted by trial number after pooling across sessions (*Figure 6A*; N = 8057 trials from 23 sessions; see Materials and methods). Interestingly, the effect on confidence (blue trace) appears to dissipate more quickly than the effect on choice, and 'over-shoots' zero for a period of time roughly between trial numbers 500–800 (~40–65 min), before settling back to zero later in the session (*Figure 6A*). Note that positive values on the ordinate indicate leftward shifts of the function, as observed previously with μStim, and are thus opposite of the expected effect from inactivation. Although the trend was not statistically significant (p=0.14; *Equation 3* applied to trial numbers 550–750), it raises the possibility of a dissociation between choice and confidence on this particular subset of trials (see Discussion). Lastly, for comparison with previous work, we performed the same analysis on the μStim data from monkey D (*Fetsch et al., 2014a*). Unlike photosuppression, and contrary to the findings of *Salzman et al. (1992)*, the biases in choice and confidence induced by μStim remained stable throughout the behavioral session (*Figure 6—figure supplement 1A*). This result implies that the capacity of cortical circuits to compensate for artificially induced activity patterns may be greater for photosuppression compared to μStim, although we cannot ascertain whether the key difference is the sign (suppression versus stimulation) or the modality (optogenetic versus electrical) of the perturbation.

The presence of compensatory changes in readout across trials led us to wonder whether such changes could occur on a faster time scale as well. We addressed this question by grouping trials

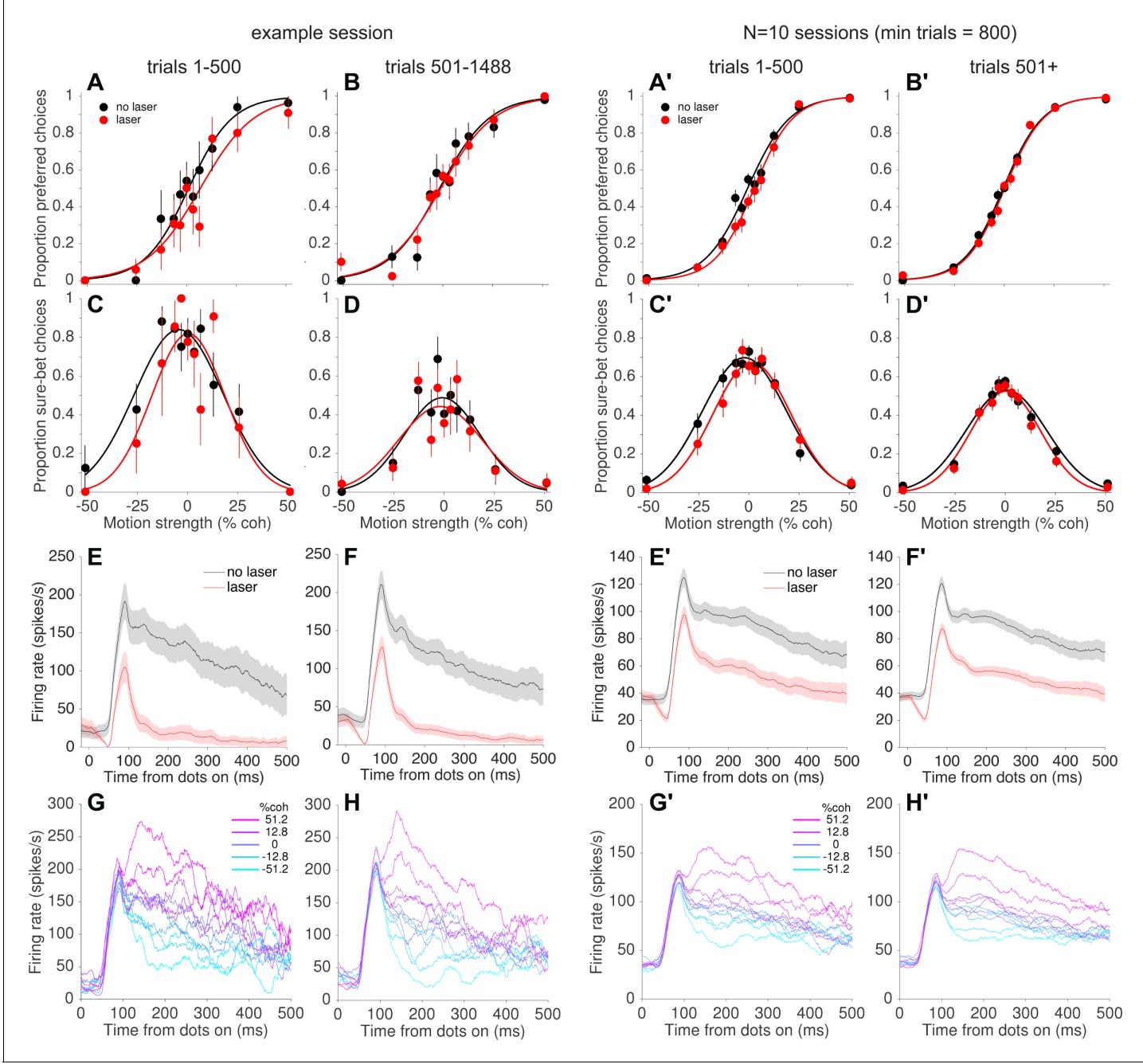

**Figure 5.** Behavioral, but not neural, effects of photosuppression dissipated over time within a session. (**A, B**) Choice functions for laser (red) and no-laser (black) trials for an example session, separated by trials early within the session (A, trials 1–500) versus later in the same session (B, trial number >500). Error bars show standard error of the proportions. (**C, D**) Corresponding confidence functions for the same groups of trials in A and B, respectively. The decrease in overall sure-bet proportion between C and D, indicating an increase in overall confidence over the course of a session, was a behavioral peculiarity of one monkey that was unrelated to photosuppression (i.e., occurred throughout training and in no-laser control sessions). (**E,F**) Average multi-unit activity showing a similar degree of suppression for early (**E**) and late (**F**) trials. Shaded regions indicate ± SEM. (**G,H**) Average firing rate on no-laser trials, early (**G**) vs. late (**H**) in the session, separated by signed motion coherence (positive = preferred direction, magenta; negative = null (antipreferred) direction, cyan). (**A'–H'**) Same as A–H but for all sessions with >800 trials (N = 10 sessions, 4438 laser trials with $\Delta R < -0.25$, 6489 no-laser trials).

DOI: https://doi.org/10.7554/eLife.36523.015

The following source data is available for figure 5:

**Source data 1.** Data and Matlab code for reproducing all panels for *Figure 5*.

DOI: https://doi.org/10.7554/eLife.36523.016

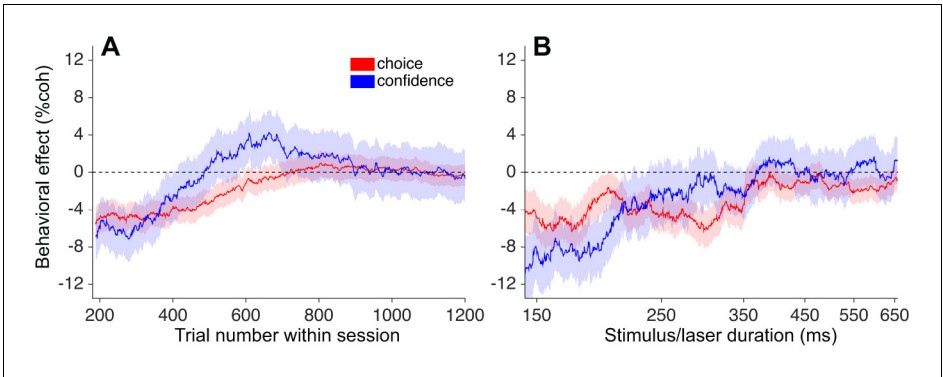

**Figure 6.** Attenuation of behavioral effects of photosuppression on long and short time scales. (**A**) Shifts of the choice (red) and confidence (blue) functions (± SEM) as a function of trial number (~8 min per 100 trials). Trials were pooled across sessions, conditioned on ΔR < −0.25 and duration <300 ms (N = 8057), and behavioral effects (***Equations 1 and 3***) were calculated over a sliding window of 1750 trials sorted by trial number. (**B**) Corresponding sliding-window analysis of trials sorted by stimulus/laser duration, after conditioning on ΔR < −0.25 and trial number <500. N = 7381 trials, window width = 1600 trials. ***Figure 6—figure supplement 1*** shows the absence of such attenuation in μStim data collected previously. ***Figure 6—figure supplement 2*** repeats these analyses after omitting one outlier session (see Materials and methods and ***Figure 4—figure supplement 3***).
DOI: https://doi.org/10.7554/eLife.36523.017

The following source data and figure supplements are available for figure 6:

**Source data 1.** Data and Matlab code for reproducing all panels and figure supplements for ***Figure 6***.
DOI: https://doi.org/10.7554/eLife.36523.020

**Figure supplement 1.** Effects of electrical microstimulation (μStim) as a function of trial number and duration.
DOI: https://doi.org/10.7554/eLife.36523.018

**Figure supplement 2.** Results of ***Figure 6A–B*** after omitting one outlier session (see Materials and methods and ***Figure 4—figure supplement 3***).
DOI: https://doi.org/10.7554/eLife.36523.019

according to the experimenter-controlled stimulus (and laser) duration, which varied randomly across trials (range = 95–925 ms, truncated exponential distribution; ***Figure 1—figure supplement 1A***). Remarkably, we found that the behavioral effects of photosuppression were largely restricted to trials with short and intermediate durations (duration <~350 ms; ***Figure 6B***). ***Figure 7*** depicts this trend pooled across all sessions, in the same format as ***Figure 5*** and limited to the first 500 trials of each session. Short-duration trials exhibited light-induced shifts of −4.7% (p=0.0006, ***Figure 7A***) and −5.7% (p=0.004, ***Figure 7C***) for choice and confidence respectively, whereas long-duration trials showed no systematic biases in either measure (choice: −0.6%, p=0.50; confidence: 0.38%, p=0.84). The effect of duration was statistically significant (***Equation 5***, Materials and methods) for confidence (p=0.002, Bonferroni corrected) and marginally so for choice (p=0.04).

As with the slower form of compensation, this sub-second attenuation of behavioral effects was not readily explained by differences in MT neural responses on short- versus long-duration trials (***Figure 7E–H***). We did observe weaker direction selectivity during the latter half of the stimulus presentation on long-duration trials (***Figure 7H***). It is conceivable that these late epochs contributed less to behavior, under the assumption that neural signals contribute in proportion to their sensitivity or informativeness (***Britten et al., 1996***; ***Gu et al., 2007***; ***Purushothaman and Bradley, 2005***). However, the relevant comparison here is between short- and long-duration trials, not early versus late within a trial. Unsurprisingly, neuronal sensitivity was greater on long-duration trials when including all spiking activity throughout the stimulus epoch (neuronal threshold = 27.2 ± 1.6% coh versus 37.5 ± 2.5% for short durations; see Materials and methods). Thus, without additional assumptions, the weaker direction selectivity at time points >300 ms (less separation of the traces on the right side of ***Figure 7H*** versus 7G) cannot explain the inability of photosuppression to affect behavior on long-duration trials. Indeed, any candidate explanation for this result must contend with the fact that activity early in the stimulus epoch was suppressed equally on short- and long-duration trials (***Figure 7E*** vs. 7F), yet only the former showed behavioral effects. It seems improbable that

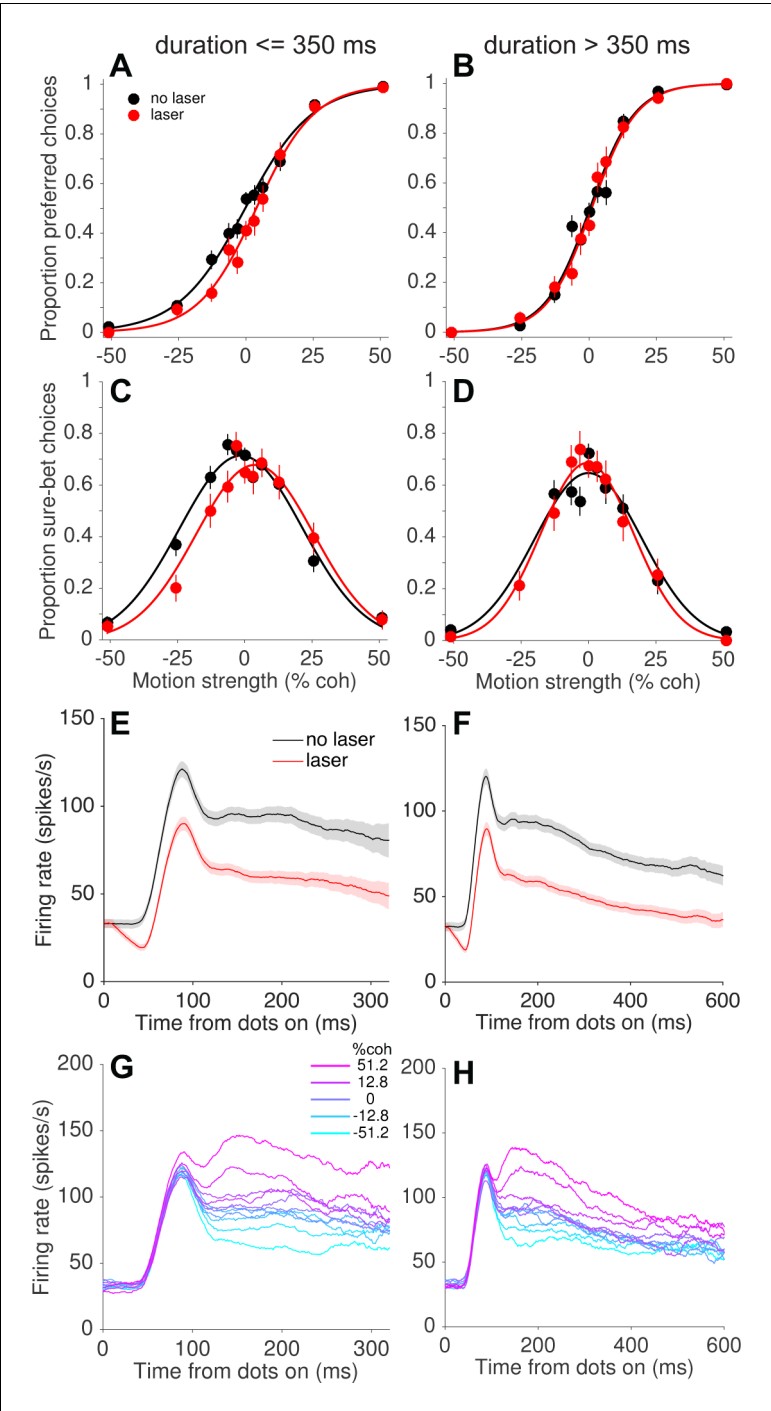

**Figure 7.** Compensation on a sub-second time scale. Behavioral effects of photosuppression were present on trials with stimulus/laser epochs of short duration but not of longer durations (left and right columns, respectively), whereas the degree of suppression and direction selectivity were similar across durations. Format as in *Figure 5*. Data include trials with $\Delta R < -0.25$ and trial number <500 (N = 4249 and 3132 trials for left and right columns, respectively). All durations were randomly interleaved.

DOI: https://doi.org/10.7554/eLife.36523.021

The following source data is available for figure 7:

**Source data 1.** Data and Matlab code for reproducing all panels for *Figure 7*.

DOI: https://doi.org/10.7554/eLife.36523.022

downstream circuits could treat such early activity differently, because at any given time during a trial there would be no way to predict whether the stimulus (and laser) would terminate in the next moment or continue. Below we discuss possible explanations and implications of this unexpected finding.

## Discussion

### Focal optogenetic suppression

The results demonstrate that systematic behavioral effects of optogenetic manipulation in nonhuman primates are achievable even when this requires the targeting of neural populations on a small spatial scale, that of cortical columns or clusters of neurons with similar functional properties. This is the scale at which electrical microstimulation has proven most effective, but nonhuman primate neuroscience has lacked an inactivation counterpart with comparable spatial and temporal specificity, relying on pharmacological and cooling methods that persist for minutes to days and affect larger swaths of tissue (but see *Afraz et al., 2015*). There is justifiable excitement surrounding the development of optogenetic methods for targeting particular cell types or anatomical projections (*El-Shamayleh et al., 2017*; *Klein et al., 2016*; *Stauffer et al., 2016*; *Ohayon, 2017*), and many questions will require such an approach. Our study shows that another form of targeting—that of spatially or topographically organized populations—is viable in monkeys using off-the-shelf viral vectors combined with careful site selection and light delivery.

### A common mechanism for choice and confidence

Decision confidence is defined as the subjective degree of belief that one has chosen the correct or more rewarding option among alternatives. In many natural situations, the decision maker does not receive immediate feedback on the accuracy of a choice, but instead must rely on confidence to guide subsequent decisions that depend on previous outcomes (*Averbeck, 2015*; *van den Berg et al., 2016b*). Confidence also has a powerful influence on learning and adapting to new environments. For instance, when negative feedback follows a choice made with high confidence, it implies that something about the world has changed, triggering a shift in behavioral strategy or an increase in learning rate (*Purcell and Kiani, 2016*; *Yu and Dayan, 2005*).

The neural basis of assigning confidence in a decision is not well understood. One class of hypotheses asserts that the processing of the evidence that supports a choice simultaneously furnishes an assessment of the reliability of the source of evidence, and hence a prediction about whether the choice will be correct (for reviews see *Fetsch et al., 2014b*; *Kepecs and Mainen, 2012*; *Meyniel et al., 2015*; *Pouget et al., 2016*; *Yeung and Summerfield, 2012*). In this vein, *Kiani and Shadlen (2009)* demonstrated how choice, reaction time, and confidence can arise through a common mechanism of evidence accumulation, and several studies have since provided additional support for this idea (*Fetsch et al., 2014a*; *Kiani et al., 2014*; *van den Berg et al., 2016a*; *Zylberberg et al., 2016*). Using the same task as in the present study, we found that the quantitative link between choice and confidence was preserved under MT/MST microstimulation (*Fetsch et al., 2014a*). Changes in both measures—and their specific relationship with motion strength and viewing duration—evinced a commensurate change in the strength of momentary evidence, and the improvement in discrimination sensitivity when confidence was high (i.e., *Figure 1C*) was retained, as predicted by a bounded evidence accumulation model.

The present study provides an inactivation counterpart to this approach. Photosuppression in MT biased the animal's choices away from the preferred direction, and also altered confidence in a way that mimics a change in motion strength (i.e., a rightward shift of the bell-shaped curve describing sure-bet choices as a function of coherence). The effects were reasonably well matched as a function of the degree of suppression (*Figure 4B*), and they dissipated over a similar time course, both across (*Figure 6A*) and within trials (*Figure 6B*). These results are consistent with a role for MT neurons in providing a shared source of evidence supporting both aspects of a decision. Some features of the data may seem, at first glance, incompatible with this straightforward account. For instance, photosuppression appears to have exerted greater effects on confidence when the motion was in the anti-preferred direction (negative coherence values, see *Figures 4D*, *5C'* and *7C*), and this asymmetry was often accompanied by a subtle increase in the slope of the choice function (most evident in

*Figure 7A*). Interestingly, these observations can be reconciled in a bounded accumulation framework, the interpretation being that photosuppression caused both a bias and an increase in discrimination sensitivity by improving the signal-to-noise, just as µStim appeared to reduce it (*Fetsch et al., 2014a*). Such a change would affect both the slope of the choice function and the width of the confidence function, the latter giving rise to the asymmetry. Our data set lacks sufficient power to conclusively support or reject this possibility, but if borne out by additional experiments, it would be within a family of previously observed alterations to signal and noise that jointly affect confidence and choice in a manner predicted by bounded accumulation (*Zylberberg et al., 2016*).

Other subtle anomalies are not as easily incorporated in this framework. The effect of photosuppression on confidence (*Figure 6A*, blue) dissipated more rapidly across trials than the effect on choices (*Figure 6A*, red), and even transiently shifted in the opposite direction (positive, or leftward). Effects at the shortest durations (*Figure 6B*) were greater for confidence than for choice (but see *Figure 6—figure supplement 1* to compare to the size of such deviations observed with µStim). Lastly, the size of the effect on choices tended to be greater on trials without the sure-bet option available, compared to when it was available (data not shown), which was not the case for µStim (*Fetsch et al., 2014b*). None of these discrepancies reached statistical significance, but taken together, they suggest the possibility that the neural substrates of choice and confidence might be partly dissociable at the level of the representation of sensory evidence. It would be of interest to deploy manipulations that alter the correspondence between choice and confidence to a greater degree than what we observed here, while recording simultaneously from sensory and higher order areas. Recent studies in humans point toward several frontal lobe structures involved in generating and reporting the sense of confidence (*Bang and Fleming, 2018*; *Fleming et al., 2015*; *Morales et al., 2018*; *Shekhar and Rahnev, 2018*). Little is known about where and how such computations occur in the nonhuman primate brain, although the supplementary eye field seems to play a role (*Middlebrooks and Sommer, 2012*; *So and Stuphorn, 2016*).

## Compensatory changes in the readout of sensory cortical activity

Behavioral effects of photosuppression were strongly attenuated over the course of the session (*Figures 5* and *6A*). This trend was not explained by differences in the physiological effectiveness of photosuppression, nor in the responsiveness or selectivity of the affected neurons. Rather, the results suggest a form of compensation or down-weighting of MT signals by downstream areas that are reading out this activity to form a decision. Because the decision in our task is rendered with a saccadic eye movement, the readout (and hence compensation) likely involves a network of areas including the lateral intraparietal area, frontal eye field, and superior colliculus, although probably by way of the thalamus rather than direct projections from MT.

What is the nature of the signal that triggers this compensation? In general, a pure choice bias will tend to increase the error rate, so a mechanism that detects and counteracts the bias could be beneficial in terms of maximizing reward. Downstream circuits could, in principle, learn to discount signals that were associated with a lower rate of reward, even though the perturbation was randomly interleaved across trials. However, in our experiments the reward rate was about 4% *greater* on laser trials compared to no-laser trials, owing to the small magnitude of the bias and the subtle increase in sensitivity. Thus, it is difficult to see how a reward-based associative mechanism could play a major role.

We think the key may lie in the faster form of compensation, revealed by the inability of photosuppression to influence behavior on long-duration trials (*Figures 6B, 7B,D*). This observation is surprising because the different durations were randomly interleaved, suggesting that the compensation must be induced by the suppression itself and initiated within a few tenths of a second. We do not know how this fast compensation comes about, but it is not a trivial consequence of the brain placing greater weight on sensory evidence arriving earlier within a trial (e.g., *Kiani et al., 2008*; *Raposo et al., 2012*), since early MT activity is suppressed to the same degree on long versus short trials (*Figure 7E,F*). Instead, the results motivate the hypothesis that the suppressed activity is sensed as anomalous or unreliable by downstream circuitry. That is, unlike µStim, the suppression of cortical tissue might have an effect that is analogous to diaschisis (*Carrera and Tononi, 2014*; *Otchy et al., 2015*), triggering a reconfiguration of functional connectivity in downstream networks. This reconfiguration could cause structures involved in decision making to consult other sources of evidence, either elsewhere in MT or from other visual areas (e.g., MST, V3a, or V4). Initially this

happens only after the first few hundred ms within a trial, then gradually the readout becomes able to rely on these signals from the start of the trial, resulting in late trials showing no effect of suppression even when limiting the analysis to short durations (*Figure 6A*). This account is admittedly speculative, but it may suggest an opportunity for filling a major gap in our understanding, namely how information is dynamically routed from a multitude of possible sources (both sensory and mnemonic; *Shadlen and Shohamy, 2016*) to the circuits responsible for decision and action.

## Conclusion

The approach taken in this study was motivated by a large body of research linking the functional properties of sensory neurons to the formation of a perceptual decision (e.g., *Afraz et al., 2006*; *DeAngelis et al., 1998*; *Romo et al., 2000*; *Salzman et al., 1992*). In nonhuman primates, the effectiveness of modern causal tools for manipulating behavior has been limited, although not without successes. One sensible response to this reality is to develop methods for illuminating larger volumes of tissue (*Acker et al., 2016*; *Gerits et al., 2012*). In some cases, however, a more nuanced understanding of the relationship between neural activity and behavior can yield important observations that would be obscured by the large-volume approach. Naturally, the ideal spatial scale depends not only on the anatomy but on the specific question being addressed. For researchers seeking to exploit the temporal control afforded by optogenetic inactivation, we suggest it is also worthwhile to give consideration to the spatial distribution of functional properties, including columnar and laminar segregation (*Chandrasekaran et al., 2017*; *Chen et al., 2017*; *Self et al., 2013*), as well as mesoscale connectivity patterns (*Lewis and Van Essen, 2000*). Relating these properties to behavior in the context of a well-characterized task can furnish insights that go beyond qualitative statements about causal involvement.

The nullification of behavioral effects of photosuppression on two distinct time scales (*Figures 5–7*) invites caution when interpreting inactivation experiments that fail to affect behavior. A key role for MT in visual motion discrimination is not in doubt (but see *Liu and Pack, 2017*), but for other areas and tasks that have not been extensively studied, or where the results are more equivocal, it may be difficult to rule out compensation by circuits that perform the same or similar function (e.g., *de Lafuente et al., 2015*). The present study suggests that this compensation can operate on a faster time scale than previously appreciated, further advocating the use of rigorous behavioral methods to identify the appropriate time scale for a perturbation. While these observations pose challenges to experimental neuroscience, they also attest to the brain's capacity for functional recovery after a small insult. One might hope that the tools for investigating causality in systems neuroscience could help expose mechanisms that mediate such recovery so that they may be exploited for neurological rehabilitation.

## Materials and methods

### Behavioral task

Two adult male rhesus monkeys (*Macaca mulatta*) performed a direction discrimination task with post-decision wagering (PDW, *Figure 1A*), as described previously (*Fetsch et al., 2014a*; *Kiani and Shadlen, 2009*; *Zylberberg et al., 2016*). In brief, animals were required to fixate a central target, after which two direction-choice targets appeared 9° to the left and right of the fixation point, followed by a dynamic random-dot motion (RDM) stimulus. Properties of the RDM (patch size, position, and dot speed) were set for each experimental session to match the aggregate receptive field (RF) and tuning properties of neurons throughout a 200–400 µm region near the optrode tip (see below for site selection criteria). The monkeys had previously performed only left-versus-right discriminations for several years and were resistant to training on up-versus-down, thus we bypassed sites with preferred directions >30° away from horizontal. Motion strength, or coherence (percent coherently moving dots), on each trial was sampled uniformly from the set {0, 3.2, 6.4, 12.8, 25.6, 51.2}, and stimulus duration was drawn from a truncated exponential distribution with range = 95–925 ms, mean = 350 ms, and median = 300 ms.

After motion offset, the monkey maintained fixation through a variable delay period during which a third target (the sure-bet target, $T_s$) was presented on a random half of trials. $T_s$ differed in color and size from the direction choice targets, and was positioned 6° above the fixation point,

perpendicular to the axis of motion and the direction-choice targets. Whether or not $T_s$ was presented, the delay period ended with disappearance of the fixation point, which served as a 'go' cue for the monkey to saccade to one of the targets. If $T_s$ was available, the monkey could choose it and receive a guaranteed reward (drop of water or juice), or waive it and make the higher-stakes direction choice; otherwise only the binary direction choice was available. Correct direction choices yielded a larger liquid reward than $T_s$ choices, while errors resulted in a 4.5 s timeout. The ratio of sure-bet reward size to direction-choice reward size (mean = 0.51 for monkey D, 0.62 for monkey N) was set and periodically adjusted to encourage the animals to choose $T_s$ approximately 60% of the time at the weakest motion strengths.

## Surgery and injection of viral vector

Procedures were in accordance with the Public Health Service Policy on Humane Care and Use of Laboratory Animals, and approved by Columbia University's Institutional Animal Care and Use Committee. Monkeys were surgically implanted with a head post and recording cylinder using aseptic technique. Cortical area MT (left hemisphere in monkey D, right in monkey N) was initially targeted using structural MRI scans and confirmed using established physiological criteria prior to injection of virus.

Viral vector injections were performed while the animals were awake and seated in a primate chair. A glass microinjection pipette (115 µm outer diameter; Thomas Recording GmbH, Giessen, Germany) was affixed to a tungsten microelectrode (75 µm shank diameter; FHC, Inc., Bowdoin, ME) with cyanoacrylate to create a custom 'injectrode' which was passed through a transdural guide tube and advanced into area MT using a hydraulic microdrive. The metal connector at the back of the pipette was coupled with flexible tubing to a Hamilton syringe loaded with 5–10 µl of virus (AAV8-CamKIIα-Jaws-KGC-GFP-ER2; titer = $5.9 \times 10^{12}$ genomes/ml; UNC Vector Core, *Chuong et al., 2014*).

After locating a stretch of gray matter with neuronal responses consistent with the functional properties of MT, we advanced the injectrode to the deepest point of this stretch and began the first of a series of injections. Using a syringe pump (Harvard Apparatus, Holliston, MA), we injected 0.75–1.0 µl at a rate of 0.05 µl/min at each of several locations spaced 400–500 µm apart. Each injection was followed by a 10 min wait period before slowly (5 µm/s) retracting the injectrode to the next site. This process continued until reaching the shallowest point of the target region, resulting in 5–8 sites and a total of 4–7 µl injected along a given track. On subsequent days, the procedure was repeated in 1 (monkey N) or 3 (monkey D) neighboring grid holes (1 mm apart) for a total injected volume of 13 and 19 µl, respectively. We then waited at least 8 weeks before beginning experiments. The tissue remained viable and responsive to light for at least 12 months post-injection in monkey D and at least 9 months in monkey N. Some virus likely reached portions of nearby area MST, but all sites tested in the behavioral task were classified as residing in MT based on published criteria (*Churchland et al., 2007*; *Komatsu and Wurtz, 1988*; *Tanaka et al., 1986*), including RF size versus eccentricity, tuning for slower dot speeds (<20°/s), and position relative to the white matter ventral to the sulcus.

To examine viral vector-mediated opsin expression histologically, we injected a third animal (monkey E) following the same procedure and using the identical vector stock injected in monkeys N and D. Monkeys N and D remain actively contributing to other projects, thus precluding histological analysis of these animals. Monkey E received a total of 6.5 µl of vector delivered at seven injection sites spaced 400 µm apart along a single penetration. These injections were made in the lateral bank of the intraparietal sulcus, rather than in MT, due to constraints imposed by an existing recording chamber.

One month after the injections, monkey E was euthanized with an overdose of pentobarbital and perfused transcardially with 4% paraformaldehyde followed by a gradient of sucrose in phosphate buffer (10%, 20% and 30%). The brain was removed and cryoprotected in 30% sucrose. Coronal sections (50 µm) were cut on a sliding microtome and mounted onto slides. Transduced cells were first localized by inspecting native fluorescence signals in a series of sections spanning the intraparietal sulcus. Additional sections near the region of strongest expression were processed immunohistochemically using primary antibodies against GFP (Abcam 13970 RRID:AB_300798, 1:1000; Abcam, Cambridge, MA) and against the pan-neuronal marker NeuN (Millipore MAB377 RRID:AB_2298772; MilliporeSigma, Burlington, MA) or the inhibitory neuron marker parvalbumin (Swant PV235 RRID:

AB_10000343 1:5000; Swant, Marly, Switzerland), and using secondary antibodies (Invitrogen Molecular Probes, Thermo Fisher Scientific, Waltham, MA): Alexa 594 (A21203 RRID:AB_141633, 1:400), Alexa 568 (A10042 RRID:AB_2534017, 1:400), Alexa 488 (custom, 1:400) and the nuclear stain DAPI (Invitrogen Molecular Probes D-21490, 1:5000) for visualization by epifluorescence microscopy. An upper bound on transduction selectivity for excitatory neurons was coarsely estimated by counting the number of GFP-positive, PV-positive and double-labeled somata in a representative section imaged at 20X (*Figure 2B,C*).

## Optrode design, site selection and photosuppression protocol

Photosuppression was achieved using a custom optrode of similar design to the injectrode described above: a tungsten microelectrode (75 or 100 μm shank diameter,~1 MOhm; FHC) glued to an optical fiber (Thorlabs UM22-100, core+cladding diameter=110 μm; Thorlabs, Newton, NJ). The fiber tip was sharpened as follows to reduce tissue damage and generate a broader light cone (*Dai et al., 2014*; *Hanks et al., 2015*): After stripping the fiber jacket, the tip was immersed in 48% hydrofluoric acid to a depth of 0.5–1.0 mm, with the polyimide coating intact to enhance smoothness and reproducibility of the resulting tip ('tube etching'; *Lambelet et al., 1998*; *Stöckle et al., 1999*). The immersion took place within the narrow end of a standard 100 μl pipette tip containing the acid and sealed with Parafilm. After 30–45 min, the acid was rinsed off and the polyimide coating mechanically stripped, leaving 6–10 mm of bare cladding and a conical tip with full angle of approximately 15°. The sharpened fiber was then passed through the guide tube along with an electrode and the two were glued together using a tiny amount of cyanoacrylate, with the electrode tip positioned 300–500 μm ahead of the fiber tip. The source end of the fiber patch cable was connected to a 633 nm diode laser (LuxX +633–100, Omicron-Laserage Laserprodukte GmbH, Rodgau, Germany) under analog and digital control.

Prior to each session, we used a handheld power meter (Thorlabs PM160) to calibrate the total light power as a function of analog input to the laser controller. Given the known values for the beam half-angle in air (20–30°) and separation between fiber and electrode tips (300–500 μm), we estimated the irradiance of neurons near the electrode tip using the brain tissue light transmission calculator provided by the Deisseroth Lab a Stanford University (https://web.stanford.edu/group/dlab/cgi-bin/graph/chart.php), as well as additional Monte Carlo simulations (*Stujenske et al., 2015*) (*Figure 3—figure supplement 1*). The goal (see below) was to use the minimum irradiance required to saturate the reduction in firing rates of neurons within a small region of interest (300–500 μm diameter), taking advantage of the fact that irradiance in tissue falls off exponentially with distance. Thus, total light power was nearly always kept below 2 mW (16 mW/mm$^2$ at a distance of 300 μm) and was typically between 0.3 and 1.0 mW (1.2–4.0 mW/mm$^2$) during the discrimination task (see *Figure 3—figure supplement 2*).

After advancing the optrode into area MT and pausing 20–40 min to allow the tissue to stabilize, we isolated single-unit (SU) or multi-unit (MU) activity using Plexon SortClient software (Plexon, Inc.). RF size, position, and selectivity for motion direction and speed were assessed using briefly flashed, 99% coherence RDM stimuli while the animal fixated a central target. Candidate sites were mapped extensively before each behavioral session to ensure consistency of tuning/RF parameters across at least 200 and preferably >300 μm of cortex. Responsiveness to red light was concurrently assessed by randomly interleaving laser trials (laser + RDM) with no-laser trials (RDM only), and sites were bypassed unless we observed a significant decrease in MU firing rate on laser versus no-laser trials (*t*-test, p<0.01) throughout the target area. A site was considered provisionally acceptable if (a) RF and tuning parameters remained relatively stable (Δ preferred direction <45°) for at least 200 μm, (b) direction selectivity was sufficiently strong, with at least two standard deviations separating preferred and null (antipreferred) direction MU responses, and (c) activity on laser trials was reduced by at least 10%. Once we encountered an acceptable site, we attempted to position the optrode at the optimal depth with respect to the above considerations, and commenced the discrimination task. Twenty-seven sites were provisionally accepted (17 in monkey D, 10 in monkey N), but 4 sites in monkey D were rejected post-hoc after failing to maintain robust suppression of MU activity throughout the session.

On a random half of discrimination trials, including T$_s$-present and T$_s$-absent trials, red light was delivered throughout the visual stimulus epoch. The laser power profile was a constant square pulse, and in a majority of sessions was terminated with a 140 ms linear ramp-down to reduce the post-

suppression burst (*Figure 3D* and inset; *Chuong et al., 2014*). Laser onset began 20 ms after stimulus onset to partially account for visual response latency while ensuring that photosuppression began before the earliest feedforward visual inputs reached MT. Onset of the ramp-down began 20 ms after visual motion offset.

The fiber jacket and optrode shank were covered with opaque material such that no laser light was visible to the animal. Nevertheless, to rule out possible nonspecific effects of illumination, including tissue heating, we performed eight control sessions in which the optrode tip was positioned either near the surface of the brain (N = 2), or in the vicinity of the dorsal superior temporal sulcus but at least 2 mm outside the viral injection site (N = 6). No behavioral effects of illumination were observed in these control sessions.

## Data analysis

All analyses were performed using custom routines in MATLAB (MathWorks, Natick, MA). We quantified the effect of photosuppression on behavioral choices using the logistic regression model:

$$P_{pref} = \left\{1 + e^{-Q}\right\}^{-1}, \ Q = \beta_0 + \beta_1 C + \beta_2 I_L + \beta_3 C I_L \tag{1}$$

Where $P_{pref}$ is the probability of a preferred-direction choice, $C$ is signed motion coherence (positive = preferred direction, negative = anti-preferred direction), and $I_L$ is an indicator variable for the presence/absence of laser illumination. The biasing effect of photosuppression is expressed in units of coherence by the ratio $\beta_2/\beta_1$, and its effect on discrimination sensitivity (slope) is captured by $\beta_3$. The coefficients were fit by maximum likelihood estimation, with standard errors (SEs) estimated as the square roots of the diagonal elements of the inverse of the Hessian matrix. SEs were used to calculate $t$-statistics and corresponding p values to test the null hypothesis that a given $\beta_i = 0$. To quantify the difference in sensitivity when $T_s$ was available versus unavailable (*Figure 1C*), we replaced $I_L$ in *Equation 1* with an indicator variable for the presence/absence of $T_s$. For display purposes, smooth curves in *Figure 4C*, *5A,B* and *7A,B* were generated using the simpler model:

$$P_{pref} = \left\{1 + e^{-Q}\right\}^{-1}, \ Q = \beta_0 + \beta_1 C \tag{2}$$

fitted separately to laser and no-laser trials. Unless otherwise indicated, plots and analyses of choices include both $T_s$-present (waived) and $T_s$-unavailable trials.

The effect of photosuppression on confidence was quantified by fitting the probability of a sure-bet choice ($P_{sb}$) with the Gaussian function:

$$P_{sb} = A e^{-(C+\mu+\delta I_L)^2/2\sigma^2} \tag{3}$$

Where $C$ is signed coherence, $I_L$ is the indicator variable for the laser, and $A$, $\mu$, $\sigma$, and $\delta$ are free parameters. The parameter $\delta$ captures the photosuppression-induced lateral shift of the Gaussian in units of coherence.

For each laser trial, we estimated the degree of suppression of neural activity as follows. Multi-unit spike events were counted from 60 ms after RDM onset until 60 ms after RDM offset, then converted to spike rate ($R_{laser}$) by dividing by the stimulus duration. The fractional change in spike rate ($\Delta R$) was then computed by comparing each $R_{laser}$ with the mean spike rate of a matching set of no-laser trials ($\bar{R}_{no-laser}$), defined as being of the same session, motion direction, coherence, and quartile of stimulus duration:

$$\Delta R = (R_{laser} - \bar{R}_{no-laser})/\bar{R}_{no-laser} \tag{4}$$

Behavioral effects as a function of $\Delta R$ (e.g., *Figure 4B*) were computed by selecting laser trials with a given range of $\Delta R$ and comparing them to all no-laser trials across all sessions, using *Equations 1 and 3* for choice and confidence, respectively. The traces in *Figure 4B* were mapped out by repeating this procedure for each of many sliding windows of trials sorted by $\Delta R$, where the window width was 2100 trials (approximately ⅕ of laser trials, i.e. a sliding quintile) and the step size was 10 trials. Similarly, *Figure 6A and B* were constructed by sorting trials by the variable on the abscissa (conditioned on two other post-hoc variables; see below) and computing behavioral effects using a sliding quartile window.

Having observed variation in the size of behavioral effects as a function of three main explanatory variables—fractional change in spike rate on laser trials (ΔR), trial number within session (T), and stimulus/laser duration (D)—we performed post-hoc statistical tests of these relationships using binomial generalized linear models (logistic regression, using the MATLAB function *glmfit*). The trends depicted in *Figure 6* were qualitatively similar in the two monkeys, thus we pooled the data for these analyses. One session showed clear shifts in the opposite direction than predicted by the local MU activity (red data point in upper-right quadrant of *Figure 4—figure supplement 3*), yet these paradoxical effects were still sensitive to trial number and duration. Since our main focus was on the time course of these trends rather than on the size of behavioral effects overall, we manually reversed the direction preference associated with this individual session before pooling with the rest of the data. Excluding this session did not alter the qualitative trends shown in *Figures 5–7* or the results of the main statistical tests.

Each post-hoc GLM consisted of three predictors, plus all 2-way interaction terms. The predictors were: signed motion coherence (C), the indicator variable for presence of the laser ($I_L$), and an indicator variable defining a median split for one of the three variables mentioned above (ΔR, D, or T). For example, the model to test the influence of stimulus duration D on photosuppression-induced choice effects was specified by:

$$P_{pref} = \left\{1 + e^{-Q}\right\}^{-1}, \; Q = \beta_0 + \beta_1 C + \beta_2 I_L + \beta_3 I_D + \beta_4 C I_L + \beta_5 C I_D + \beta_6 I_L I_D \tag{5}$$

Where $I_D = 1$ if D < 300 ms (the median duration) and 0 otherwise. The model for ΔR substituted $I_{\Delta R}$ (assigned to 1 if ΔR < -0.35, the median ΔR) in place of $I_D$, and for trial number the corresponding variable was $I_T$ (1 if T < 578, the median trial number). Since the relationship between trial number and the confidence effect (*Figure 6A*, blue) was found to be non-monotonic, we used a lower cutoff value of T < 300 to define $I_T$ for that test. For each model, the strength of the interaction between the given explanatory variable and the biasing effect of photosuppression is captured by $\beta_6$ and its associated p value.

A similar strategy was used for the effects on confidence (replacing $P_{pref}$ with $P_{sb}$ in *Equation 5* and its counterparts for ΔR and T), except that the models were fit separately for trials with C < 0 and C > 0. This piecewise-logistic approach achieved adequate fits because each side of the bell-shaped relationship between $P_{sb}$ and C is well approximated by a sigmoid, albeit of opposite slope. This yielded two $\beta_6$ coefficients and corresponding p values, and the test was deemed significant if either were less than 0.05 after Bonferroni correction (multiplication by 2). To better isolate the effects of trial number, all GLM analyses, as well as the sliding-window plot in *Figure 6A*, included only trials with ΔR < −0.25 and D < 300 ms. Similarly, the GLMs for duration (and *Figures 6B*, *7A–D*) were limited to only trials with ΔR < −0.25 and T < 500.

To compare neuronal direction selectivity across different subsets of trials—e.g., short vs. long trials (*Figure 7*)—we used a standard approach to calculate neural thresholds (*Britten et al., 1992*). For each trial in a given session, the MU spike rate was normalized to the mean spike rate for 51.2% coherence motion in the preferred direction during that session. Normalized spike rates were then pooled across sessions, grouped according to desired criteria (duration, trial number within session, etc.), and the distributions of spike rates for preferred- vs. antipreferred direction of motion were compared for a given coherence level using ROC analysis. This quantified the ability of an ideal observer to discriminate motion direction based on the neural responses. The performance of the ideal observer (percent correct as a function of coherence) was fit with a cumulative Weibull distribution (*Quick, 1974*) and the $\alpha$ parameter (coherence level generating 82% correct) was taken as the neuronal threshold.

## Acknowledgements

This research was supported by the Howard Hughes Medical Institute and the National Eye Institute. DJ is supported by a postdoctoral fellowship from the Simons Collaboration on the Global Brain. We thank Dr. Mehrdad Jazayeri (MIT) for contributions to the early development of the project, Dr. Ed Boyden (MIT) and his lab for developing and sharing the Jaws construct, Brian Madeira for technical support and animal care, and members of the Shadlen lab for discussions.

# Additional information

## Funding

| Funder | Grant reference number | Author |
|---|---|---|
| Howard Hughes Medical Institute | | Christopher R Fetsch<br>Naomi N Odean<br>Danique Jeurissen<br>Michael N Shadlen |
| National Eye Institute | EY11378 | Christopher R Fetsch<br>Naomi N Odean<br>Danique Jeurissen<br>Michael N Shadlen |
| National Eye Institute | EY018849 | Yasmine El-Shamayleh<br>Gregory D Horwitz |
| Simons Foundation | Simons Collaboration on the Global Brain Postdoctoral Fellowship | Danique Jeurissen |

The funders had no role in study design, data collection and interpretation, or the decision to submit the work for publication.

## Author contributions

Christopher R Fetsch, Conceptualization, Data curation, Formal analysis, Validation, Investigation, Visualization, Methodology, Writing—original draft, Writing—review and editing; Naomi N Odean, Conceptualization, Investigation, Methodology, Writing—original draft, Writing—review and editing; Danique Jeurissen, Investigation, Writing—original draft, Writing—review and editing; Yasmine El-Shamayleh, Investigation, Methodology, Writing—review and editing; Gregory D Horwitz, Resources, Supervision, Funding acquisition, Writing—review and editing; Michael N Shadlen, Conceptualization, Resources, Formal analysis, Supervision, Funding acquisition, Methodology, Writing—original draft, Writing—review and editing

## Author ORCIDs

Christopher R Fetsch http://orcid.org/0000-0002-7921-8306
Danique Jeurissen http://orcid.org/0000-0003-3835-5977
Yasmine El-Shamayleh http://orcid.org/0000-0002-5396-2823
Gregory D Horwitz http://orcid.org/0000-0001-5130-5259
Michael N Shadlen http://orcid.org/0000-0002-2002-2210

## Ethics

Animal experimentation: This study was performed in strict accordance with the Public Health Service Policy on Humane Care and Use of Laboratory Animals. The animals were handled according to an approved institutional animal care and use committee (IACUC) protocol (#AAAN4900) of Columbia University. Surgery was performed under isoflurane anesthesia in aseptic conditions, and every effort was made to minimize suffering.

## Decision letter and Author response

Decision letter https://doi.org/10.7554/eLife.36523.028
Author response https://doi.org/10.7554/eLife.36523.029

# Additional files

## Data availability

Matlab code (m-files) and data (.mat files) have been provided for Figure 1 and Figures 3-7.

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
