## [Decision Letter]

Thank you for submitting your article "Focal optogenetic suppression in macaque area MT biases direction discrimination and decision confidence, but only transiently" for consideration by *eLife*. Your article has been reviewed by three peer reviewers, one of whom is a member of our Board of Reviewing Editors, and the evaluation has been overseen by David Van Essen as the Senior Editor. The following individual involved in review of your submission has agreed to reveal their identity: Richard Born (Reviewer #2).

The reviewers have discussed the reviews with one another and the Reviewing Editor has drafted this decision to help you prepare a revised submission.

The reviewers agree that this is an outstanding study that makes use of a powerful visual discrimination task in which choice and confidence decision mechanisms that can be linked to the directionality of the preferred response of MT neurons. The reviewers agree that your suppression optogenetic approach in the preferred direction related MT neurons and its effects on choice and confidence constitutes an important advance to linking neural activity and behavior. Having said this, there are some comments by reviewers #2 and 3 and to a lesser extent by reviewer #1 that must be considered to improve/strengthen your paper:

1) It is not clear whether choice and confidence relate to the same decision variable, at least for the photo suppression results provided here (reviewer #2).

2) It would be useful to know whether your weak effects obtained on both choice and confidence were due to the low light power used; the reviewers wonder whether you used higher light power in other sessions (reviewers #2 and 3). This could show that the effects are more comparable to those reported before with your electrical micro-stimulation method. This might also shed light on the spatial extent of suppression by light and its power dependence (reviewers #2 and 3).

3) Please comment on whether you tried up-down discriminations while inactivating right-left preferred columns as a control (reviewers #2 and 3).

4) The short and long temporal suppressive effects on decisions were pooled from sessions in the two monkeys; the reviewers wonder about the behavioral performances of each animal.

Reviewer #1:

This study is technically well executed. This research group previously proved that electrical micro-stimulation of MT increased the speed of decisions in favor of the preferred direction while slowing decisions in favor of the opposite directions. However and according to the authors, there are technical limitations using the micro-stimulation technique, and therefore the need of some other ways of perturbing clusters of neurons associated with behavior. Here, they used an optogenetic approach in which they suppressed small clusters of excitatory neurons with selectivity for visual motion. They found that optogenetic suppression of small clusters of MT neurons induced a weak choice bias against the neurons' preferred direction, consistent with a common mechanism underlying choice and confidence. These weak effects on choice and confidence were supposedly due to compensatory changes in the read out of MT activity. A more intriguing result was that these weak changes were present only during the first third of the perturbing session (first 500 trials). This was consistently observed across many sessions. Importantly, in all the recorded neuronal sessions, single and multi-unit activity decreased during the optogenetic test, showing the reliability of this suppression approach. As said above, the paper is technically well executed and the experiment is well designed. Of course, one major concern relies on the suppression effects of MT activity and therefore on the behavioral results. The experiment is fair since it shows the opposite of what a micro-stimulation effect was in this task. I understand the authors for using new experimental tools, and appreciate the finesse of the statistical analyses and the explanation they gave on the temporal suppression effects on behavior. I should say that this paper is one of the finest ones that I have seen using the optogenetic tool in behavior. But, honestly, I did not learn much of this study.

Reviewer #2:

The transfer of optogenetic (and other virally or genetically targeted) methods to the nonhuman primate is a big deal, and the authors have made several important contributions to this endeavor. First, and most importantly, they tested the new method in the context of a behavioral task that is extremely well characterized at the anatomical, physiological, behavioral and theoretical levels. Moreover, they used a slight variation of this task ("post-decision wagering") that buys them better interpretability of any effects produced. The effects they observed, while relatively modest in size by microstim criteria, are convincing and highly intriguing. They go well beyond demonstrating the efficacy of the method to reveal new and interesting results, particularly with respect to the apparent downstream compensation for inactivation at both short (within trial) and longer (across trials) time scales. A critical advance over the microstim experiments is that, here, the authors have a trial-by-trial measure of the efficacy of their inactivation. They also reveal what may be subtle differences (subject to limitations of the data, to which the authors make appropriate emphasis – subsection “Compensation on slow time scales”, last paragraph) in circuit mechanisms for choice vs. confidence. Overall, I think this is an outstanding paper that makes major methodological and conceptual contributions to our understanding of sensory decision making.

The biggest negative, and it is a relatively minor one, was the uncertainty in the spatial extent of the inactivation. In the main text, the authors supply us with only a rather vague description: "a relatively small cluster." They actually do much better in the second paragraph of the Materials and methods subsection “Optrode design, site selection and photosuppression protocol”. I suggest that some version of these lines – especially the point about confirming the lack of effect at 1 mm distances-be moved to (or repeated in) the main text. Obviously, the directional selectivity of the effects is also evidence for columnar specificity, but the authors only ever assessed the one (i.e. right-left) axis. It would be interesting to know if they ever tried the orthogonal axis (i.e. up-down when inactivating right-preferring neurons) as a negative control, or did some of the other specificity checks that have been previously performed for microstimulation: e.g. finding a lack of effect when moving the discriminand away from the recorded/inactivated receptive fields; increasing laser power to inactivate multiple direction columns to produce a flattening of the psychometric function. These are all obvious directions for future experiments and are absolutely not required for this study to have a large impact.

One other general concern: it appears that data from the two monkeys was pooled for all analyses. This is understandable. But I was not able to find a statement declaring a justification for this. Were the main results consistent between the two monkeys (except for the one-monkey peculiarity mentioned in the legend to Figure 5)?

Reviewer #3:

This paper uses optogenetic suppression of cortical activity to investigate how visual motion signals in area MT are read out during perceptual judgments. The experimental design is based on that used previously with electrical microstimulation – monkeys viewed a random dot visual motion display of variable coherence and reported their left/right judgment by making a saccade to the appropriate target and were also sometimes given a third opt-out option that gave a guaranteed smaller reward. The opt-out option provides a measure of confidence (higher opt-out rate implies lower decision confidence) and previous microstimulation results suggested that a common mechanism contributed to both confidence and choice. The major difference in this study is the use of focal optogenetic suppression that was orchestrated so that the causal effects were mostly limited to a population of MT neurons with similar directional tuning, which was confirmed with electrophysiological recordings. The ability to reversibly suppress neuronal activity with this spatial and temporal precision is a substantial and elegant advance over previous techniques.

The effects reported in the Results are subtle but clear – both choices and confidence are systematically altered by the optogenetic suppression. But unlike the case with electrical microstimulation, the causal effects are subject to some form of compensatory change, such that the effects on choice behavior are present only for the first 100 or so trials within each session and only for visual motion stimuli that are presented for shorter durations. The reasons for these compensatory changes are not known and lie beyond the scope of this paper, although the authors outline some reasonable explanations to be tested elsewhere. The main strength of the paper lies in its impressive demonstration of methodology for applying optogenetics to pose questions that cannot be addressed with older traditional techniques (e.g., stimulation, drug injection) that lack the necessary temporal and spatial precision. The results also point to some interesting distinctions between choice and confidence that are unexpected, and it seems that the authors themselves are somewhat unsure whether to fully embrace these effects, since they run counter to their previous interpretations.

Overall, this is a very well-written paper that presents novel results on a very interesting topic. My comments are mainly aimed at clarifying the results and interpretation.

The data in the paper provide a basis for possibly drawing novel distinctions between the mechanisms of choice versus confidence. The authors acknowledge that choice and confidence are not always coupled in their results, but nonetheless conclude that their data are consistent with a common mechanism for both (Abstract). As they point out, their analysis in Figure 6 shows that confidence and choice have different effects as a function of trial number. They also report that the shifts in the choice and confidence functions were not correlated, although this negative result may be hard to interpret. More to the point, when I look at the confidence curves in Figures 4D, 5A, etc., I see a very different pattern than the obvious shift that is observed during electrical stimulation (e.g., Figure 4—figure supplement 1B). Rather than a shift, it looks like optogenetic suppression results in a 1-sided (uni-directional) increase in confidence (i.e., a drop in opt-out choices for low signal strengths when the opposite motion direction is suppressed). This makes me wonder about the assumption that confidence is based on the same decision variable as the choice – that is, the difference between rightward and leftward motion signals. Is it possible that confidence is influenced by the total amount of rightward and leftward motion, and not just the magnitude of the difference? To be clear, I don't expect this issue can be settled with this data set. But there does seem to be an incongruity between the data presented and the conclusion about a 'common mechanism for choice and confidence' that should be resolved.

As part of the experimental design, very low light powers were used in order to target local clusters of MT neurons with similar selectivity. This explains why the effect sizes are modest and may not reach significance on individual sessions. For comparison with other studies, have you tried using higher light powers and illuminating larger volumes of tissue? I would expect to see stronger disruptions of performance, but for all directions of visual motion. Nonetheless, the disruptions would be specific for visual motion stimuli placed at the appropriate retinotopic locations. This might serve as a useful positive control and point of comparison with other studies.

The histology on the Jaws expression is an important aspect of the results. An obvious concern is that the histology was done in area LIP not area MT, and it is not obvious to me whether the expression patterns in area MT would be expected to closely follow those in LIP or whether they might end up being very different. Are there neuronal cell type similarities (or differences) between MT and LIP that should be pointed out when interpreting these results?

Unless I missed it, the paper presents the effects of photosuppression only for the trials that included the opt-out option. What effects do you find if the opt-out is not available? Does this substantially change the size of the effect?

---

## [Author Response]

The reviewers agree that this is an outstanding study that makes use of a powerful visual discrimination task in which choice and confidence decision mechanisms that can be linked to the directionality of the preferred response of MT neurons. The reviewers agree that your suppression optogenetic approach in the preferred direction related MT neurons and its effects on choice and confidence constitutes an important advance to linking neural activity and behavior. Having said this, there are some comments by reviewers #2 and 3 and to a lesser extent by reviewer #1 that must be considered to improve/strengthen your paper:1) It is not clear whether choice and confidence relate to the same decision variable, at least for the photo suppression results provided here (reviewer #2).

We agree this question was left somewhat unresolved, and we have now attempted to address it more fully through modeling and changes to the text. These additions clarify what we can and cannot say about this issue with the current dataset. The upshot is that we have maintained, although softened, our conclusion that the results support a common decision variable for choice and confidence. For details please see response to reviewer 3 and Author response image 4.

2) It would be useful to know whether your weak effects obtained on both choice and confidence were due to the low light power used; the reviewers wonder whether you used higher light power in other sessions (reviewers #2 and 3). This could show that the effects are more comparable to those reported before with your electrical micro-stimulation method. This might also shed light on the spatial extent of suppression by light and its power dependence (reviewers #2 and 3).

We suspect there is a fundamental limit to the size of direction-specific biasing effects that can be caused by photosuppression, for three main reasons: (i) a nontrivial proportion of neurons in the injected region fail to express the opsin, a limitation of current viral vector technology; (ii) increasing the laser power much beyond the levels we used would suppress nearby columns with different direction preferences, thus diluting the direction specificity of the effect; and (iii) the magnitude of the change in firing rate with suppression is inherently limited by floor effects, more so than microstimulation is limited by ceiling effects. We did not have the opportunity to examine the question empirically, e.g., by systematically varying light power for a given behavioral session. We did modify the text (moving some from Materials and methods to Results) and added two figure supplements to emphasize pilot experiments addressing the spatial extent and power dependence of photosuppression, based on a suggestion from reviewer 2. See Author response image 2 and new figure supplements associated with main text Figure 3.

3) Please comment on whether you tried up-down discriminations while inactivating right-left preferred columns as a control (reviewers #2 and 3).

We thank the reviewers for this idea and we hope to pursue such experiments in the future. Here we were unable to test up-down discriminations, owing to a limitation in the behavioral capacities of the particular animals used in this study. This limitation is now mentioned in Materials and methods (subsection “Behavioral task”, first paragraph); see also response to reviewer 2 below.

4) The short and long temporal suppressive effects on decisions were pooled from sessions in the two monkeys; the reviewers wonder about the behavioral performances of each animal.

The effects of suppression on early/short trials (Figure 5A’+C’, Figure 7A+C) were highly significant in each animal individually, and the trends depicted in Figure 6 were qualitatively similar in both monkeys (Author response image 3), although with some minor differences. See response to reviewer 2 below for details.

Lastly, we corrected an error in the handling of data from one of the 23 inactivation sessions. This session showed clear behavioral effects on choice and confidence, but in the opposite direction than predicted by the recorded multiunit activity (see data point in the upper-right quadrant of Figure 4—figure supplement 3, now highlighted in red). Such ‘paradoxical’ effects occasionally happen with electrical µStim as well (e.g. Salzman et al., 1992; DeAngelis et al., 1998; Fetsch et al., 2014) and are believed to imply that a population of oppositely tuned neurons – nearby but not being picked up by the recording electrode – were affected by the manipulation and contributed more strongly to behavior than the neurons being recorded (Salzman et al., 1992).

Interestingly, the effects in this individual session were sensitive to trial number (Author response image 1), duration (E-H), and degree of suppression (not shown), consistent with the patterns described for the full dataset. As the study’s emphasis shifted to the surprising compensatory effects, we decided to include this paradoxical site in these analyses. To do so required reversing the assignment of preferred and null directions, such that the sign of the behavioral shifts was negative (rightward); otherwise the compensation would have canceled out similar trends in other ‘non-paradoxical’ sessions. However, we mistakenly included the reversed session in the analyses of Figure 4, which has nothing to do with compensation. We have now remedied this by explaining the above reasoning in the Materials and methods (subsection “Data analysis”, last paragraph) and replacing Figure 4 with one that does not include the paradoxical session. We have also added a figure supplement (Figure 6—figure supplement 2) showing that the main results hold when the paradoxical session is omitted.

**Author response image 1. respfig1:** Behavioral effects of photosuppression in one ‘paradoxical’ session, separated by: early trials (A-B) versus late trials (C-D); and short durations (E-F) versus long durations (G-H).

Reviewer #1:This study is technically well executed. This research group previously proved that electrical micro-stimulation of MT increased the speed of decisions in favor of the preferred direction while slowing decisions in favor of the opposite directions. However and according to the authors, there are technical limitations using the micro-stimulation technique, and therefore the need of some other ways of perturbing clusters of neurons associated with behavior. Here, they used an optogenetic approach in which they suppressed small clusters of excitatory neurons with selectivity for visual motion. They found that optogenetic suppression of small clusters of MT neurons induced a weak choice bias against the neurons' preferred direction, consistent with a common mechanism underlying choice and confidence. These weak effects on choice and confidence were supposedly due to compensatory changes in the read out of MT activity. A more intriguing result was that these weak changes were present only during the first third of the perturbing session (first 500 trials). This was consistently observed across many sessions. Importantly, in all the recorded neuronal sessions, single and multi-unit activity decreased during the optogenetic test, showing the reliability of this suppression approach. As said above, the paper is technically well executed and the experiment is well designed. Of course, one major concern relies on the suppression effects of MT activity and therefore on the behavioral results. The experiment is fair since it shows the opposite of what a micro-stimulation effect was in this task. I understand the authors for using new experimental tools, and appreciate the finesse of the statistical analyses and the explanation they gave on the temporal suppression effects on behavior. I should say that this paper is one of the finest ones that I have seen using the optogenetic tool in behavior. But, honestly, I did not learn much of this study.

We thank the reviewer for this concise summary of our findings and the positive assessment overall. We do not understand the major concern mentioned in the sentence beginning “Of course, one major concern relies on…” We assume the final remark reflects the reviewer’s acceptance of the conclusions drawn from previous studies, especially our electrical µStim study (Fetsch et al., 2014). Nevertheless, we feel it is not a foregone conclusion that inactivating a population of neurons would have clear, interpretable effects in the opposite direction as activating them. Or perhaps the reviewer feels this study does not represent a major technical advance. We would agree, in the sense that optogenetics in nonhuman primates still has yet to reach its full potential, but hope the reviewer would concede that this is nevertheless an important step.

Reviewer #2:[…] The biggest negative, and it is a relatively minor one, was the uncertainty in the spatial extent of the inactivation. In the main text, the authors supply us with only a rather vague description: "a relatively small cluster." They actually do much better in the second paragraph of the Materials and methods subsection “Optrode design, site selection and photosuppression protocol”. I suggest that some version of these lines – especially the point about confirming the lack of effect at 1 mm distances-be moved to (or repeated in) the main text.

We agree that the issue of spatial extent was not given sufficient emphasis or clarity in the main text, and have now addressed this in several ways. First, as suggested, we reexamined the pilot sessions with 1 mm separation, but found that we had not tested high enough power levels, leaving it unclear whether the lack of an effect was because the light did not reach the neurons near the electrode tip or whether those neurons had simply failed to express the opsin. We do have some relevant data from the handful of V-probe + fiber sessions alluded to in the Materials and methods, now shown in Author response image 2, but we do not consider this direct estimate of light spread to be reliable enough (N=3 sessions) to warrant inclusion in the main text. We also provide a figure showing multiunit activity during a subset of mapping (fixation) blocks in which light power was systematically varied prior to beginning the behavioral task. This has been added to manuscript as Figure 3—figure supplement 2. Lastly, we performed new Monte Carlo simulations of light absorption/scattering in brain tissue (Figure 3—figure supplement 1), based on code made available by Stujenske et al. 2015. Each of these analyses provides support for our claim that photosuppression was typically limited to a region 300-500 µm across.

**Author response image 2. respfig2:** Qualitative estimate of the spatial extent of photosuppression, furnished by multiunit recordings during pilot mapping sessions using a 16-channel linear array (V-Probe, Plexon Inc.; N=3 sessions). Left panel: The optical fiber was glued to the probe at a known position near one of the central contacts, which were spaced 150 µm apart. Right panel: Fractional change in firing rate (+/- SEM) as a function of distance from the fiber tip (N=3 sessions). Total power for these sessions was 0.6-0.9 mW, eliciting suppression that was limited to a distance of 300 µm (although these neurons were less light-sensitive than we typically observed).

Obviously, the directional selectivity of the effects is also evidence for columnar specificity, but the authors only ever assessed the one (i.e. right-left) axis. It would be interesting to know if they ever tried the orthogonal axis (i.e. up-down when inactivating right-preferring neurons) as a negative control, or did some of the other specificity checks that have been previously performed for microstimulation: e.g. finding a lack of effect when moving the discriminand away from the recorded/inactivated receptive fields; increasing laser power to inactivate multiple direction columns to produce a flattening of the psychometric function. These are all obvious directions for future experiments and are absolutely not required for this study to have a large impact.

These are all excellent suggestions and on the docket for future experiments. We do know that microstimulation of columns with preferred directions orthogonal to the discriminanda can affect choices in a systematic way (Salzman and Newsome, 1994; Groh et al., 1997; Nichols and Newsome 2002). These effects have been interpreted as favoring either a winner-take-all or vector-averaging readout mechanism, depending on the angular separation and the specifics of the task. Unfortunately we were unable to perform this type of experiment here because the particular monkeys we used were not trained to perform up-down discriminations, and in fact were unusually resistant to such training. We have clarified the text in Materials and methods that mentions this fact (subsection “Behavioral task”, first paragraph). We also did not try placing the dots stimulus outside of the receptive field. We know this eliminates the effect of microstimulation on choices (Salzman et al., 1992), but it would be interesting to look for effects on confidence. Regarding a high-power manipulation to flatten the psychometric function, please see response to reviewer 3 below.

One other general concern: it appears that data from the two monkeys was pooled for all analyses. This is understandable. But I was not able to find a statement declaring a justification for this. Were the main results consistent between the two monkeys (except for the one-monkey peculiarity mentioned in the legend to Figure 5)?

Yes, the main results were consistent between the two monkeys: the effects of suppression on early/short trials (Figure 5A’+C’, Figure 7A+C) were significant in each animal individually, and the time course of attenuation of these effects across trials and time within a trial were qualitatively similar, as shown in Author response image 3. All four trends were statistically significant in monkey D (blue and red traces in Author response image 3), but not monkey N (Author response image 3). In panel D the trends are there, but weaker (p = 0.14 for both). In panel C they are nonsignificant because the data exhibited non-monotonicity (e.g., blue trace for trial numbers ~950-1400; see inset), whereas the statistical approach we used assumes a monotonic relationship. We do not have an explanation for the transient ‘rebound’ of the confidence effect on late trials in monkey N (inset), but it is worth noting that only a minority of sessions (~7) lasted long enough to contribute to these late data points. Overall, we felt the results were similar enough to justify pooling the two monkeys, especially given the constraints on sample size owing to technical challenges involved in acquiring the data. But we agree this should have been mentioned in the text, and now do so (Materials and methods, subsection “Data analysis”, fourth paragraph).

**Author response image 3. respfig3:** Behavioral effects of photosuppression as a function of trial number within session (left column) and stimulus/laser duration (right column), as in main text Figure 6, plotted separately for the two monkeys. Note that the abscissa range in panel C was chosen to facilitate comparison with panel A; the inset shows the full extent of trial numbers for monkey N.

Reviewer #3:[…] The effects reported in the Results are subtle but clear – both choices and confidence are systematically altered by the optogenetic suppression. But unlike the case with electrical microstimulation, the causal effects are subject to some form of compensatory change, such that the effects on choice behavior are present only for the first 100 or so trials within each session and only for visual motion stimuli that are presented for shorter durations. The reasons for these compensatory changes are not known and lie beyond the scope of this paper, although the authors outline some reasonable explanations to be tested elsewhere. The main strength of the paper lies in its impressive demonstration of methodology for applying optogenetics to pose questions that cannot be addressed with older traditional techniques (e.g., stimulation, drug injection) that lack the necessary temporal and spatial precision. The results also point to some interesting distinctions between choice and confidence that are unexpected, and it seems that the authors themselves are somewhat unsure whether to fully embrace these effects, since they run counter to their previous interpretations.Overall, this is a very well-written paper that presents novel results on a very interesting topic. My comments are mainly aimed at clarifying the results and interpretation.

We are grateful for these remarks, and agree there was some ambiguity in the interpretation relating to whether a common mechanism underlies choice and confidence. We have addressed this by fitting the data to a bounded evidence accumulation model (see below) and by improving the clarity with which we address this issue in the text.

The data in the paper provide a basis for possibly drawing novel distinctions between the mechanisms of choice versus confidence. The authors acknowledge that choice and confidence are not always coupled in their results, but nonetheless conclude that their data are consistent with a common mechanism for both (Abstract). As they point out, their analysis in Figure 6 shows that confidence and choice have different effects as a function of trial number. They also report that the shifts in the choice and confidence functions were not correlated, although this negative result may be hard to interpret. More to the point, when I look at the confidence curves in Figures 4D, 5A, etc., I see a very different pattern than the obvious shift that is observed during electrical stimulation (e.g., Figure 4—figure supplement 1B). Rather than a shift, it looks like optogenetic suppression results in a 1-sided (uni-directional) increase in confidence (i.e., a drop in opt-out choices for low signal strengths when the opposite motion direction is suppressed). This makes me wonder about the assumption that confidence is based on the same decision variable as the choice – that is, the difference between rightward and leftward motion signals. Is it possible that confidence is influenced by the total amount of rightward and leftward motion, and not just the magnitude of the difference? To be clear, I don't expect this issue can be settled with this data set. But there does seem to be an incongruity between the data presented and the conclusion about a 'common mechanism for choice and confidence' that should be resolved.

We thank the reviewer for these insightful comments. We hope the discussion below and the changes to the manuscript help resolve the incongruity between our conclusion of a common mechanism and the apparent discrepancies that the reviewer perceptively points out.

Regarding the asymmetry in the confidence function, we agree that the data seem to indicate a greater effect on confidence when the motion was in the null direction (negative coherence) compared to the preferred direction (positive coherence). We tried a couple different ways to quantify this observation (The first method was to calculate the odds ratio (OR) for the contingency table made up of the occurrence (or not) of a sure-bet choice and the presence/absence of the laser. We did this separately for each side of the confidence function (coherence > 3.2% versus < -3.2%) and computed 95% confidence intervals for the log-OR (method of Woolf). The 95% CI for positive coh was highly overlapping the abs(CI) for negative coh, meaning the asymmetry was not statistically significant. The second method was to use the width parameter of the fitted Gaussian, using a modification of Equation 3 with an indicator term applied to σ. A shift plus a decrease in width on laser trials would generate the asymmetry, but the width change was also not statistically significant.), and for both approaches it did not reach statistical significance. However, even if the asymmetry were significant, whether it entails a distinct decision variable (or mechanism) for confidence is unclear. First, we confess an inability to connect the dots between the observation and the interesting proposal the reviewer makes about confidence being influenced by the total amount of rightward and leftward motion. Trials with greater or lesser total motion could arise by chance, but were equally likely to occur for positive and negative coherence, and thus cannot straightforwardly explain the asymmetry (we did not systematically vary the total motion energy within a given coherence level). Still, we do not reject the possibility of such a mechanism, and it would be interesting to test other types of stimuli that independently vary the strength of evidence supporting each option (e.g., Zylberberg, Barttfeld, and Sigman, 2012; Odegaard et al., PNAS 2018).

It should be noted that the asymmetry could arise under a common mechanism, if photosuppression were to cause both a shift and a decrease in the width of the bell-shaped confidence function. In a bounded accumulation model (e.g., Fetsch et al., 2014), narrowing of the confidence curve is captured by an increase in the sensitivity parameter *k*, which converts the (experimenter-controlled) motion strength to the drift rate of the decision variable. Naturally, increasing *k* also entails an increase in the sensitivity, or slope, of the choice function, and indeed there is a hint of this in the data, especially in Figure 7A (although it too does not reach significance; p = 0.10, logistic regression).

To illustrate this point, we fit the model to the data shown in main text Figures 5A’+C’ and 7A+C. The fits are shown below in Author response image 4, showing that the model can account for the asymmetry in the confidence effect despite having a single decision variable that governs choice and confidence. Our sample size does not allow for rigorous model comparison, so we cannot make strong conclusions about whether and how photosuppression alters properties of sensory signal and noise. But the fitting exercise suggests that the observation of asymmetry is not impetus for rejecting the hypothesis of a common mechanism. We have now included this point in the Discussion (subsection “A common mechanism for choice and confidence”, third paragraph).

**Author response image 4. respfig4:** Fits to a bounded accumulation model (Fetsch et al., 2014) using the data shown in main text Figure 5A'+C' (left column) and Figure 7A+C (right column). The model implements the biasing effect of photosuppression – on both choice and confidence – as an offset to the mean of the momentary evidence (i.e., motion strength). It also has free parameters that permit a change in the drift rate (sensitivity) and/or the sensory noise on laser trials. These parameters can account for more subtle effects including changes in the slope of the choice function, and the height and width of the confidence function. Importantly, however, the mechanism for establishing confidence still operates on the same decision variable as the one used for choice.

Regarding the discrepancies in time course (Figure 6), the current dataset cannot rule out that they occurred by chance. Using the standard errors of the shift parameters (β2 in Equation 1, and δ in Equation 3) – and conceding that the lateral shift is only one of several aspects of the data one could examine – none of the differences between the shifts in choice and confidence rise to the level of statistical significance (see reply to comment below about the ‘overshoot’).

Nevertheless, in light of the reviewer’s concern (which we share), we have walked back some of the emphasis in the manuscript on a common mechanism for choice and confidence. For instance, we removed a sentence relating to Figure 4B: "Note that the effects (or lack thereof) on choice and confidence were reasonably well matched across the full range of ∆R (Figure 4B), supporting the idea of a common decision mechanism linking the two measures," added phrases like “broadly consistent”, and rewrote a large part of the Discussion (subsection “A common mechanism for choice and confidence”) to include several of the considerations described above.

As part of the experimental design, very low light powers were used in order to target local clusters of MT neurons with similar selectivity. This explains why the effect sizes are modest and may not reach significance on individual sessions. For comparison with other studies, have you tried using higher light powers and illuminating larger volumes of tissue? I would expect to see stronger disruptions of performance, but for all directions of visual motion. Nonetheless, the disruptions would be specific for visual motion stimuli placed at the appropriate retinotopic locations. This might serve as a useful positive control and point of comparison with other studies.

We appreciate these suggestions, although in our view it is unclear whether low light power explains the small effect sizes. Increasing laser power might have suppressed the targeted column more strongly, but this would come at the cost of affecting nearby columns/clusters and thus diluting the directional specificity of the manipulation. It is conceivable that, if laser power were to be steadily increased, one might see larger bias effects kick in before being swamped by the overall disruption in performance (Murasugi et al., 1993). We suspect that this would not occur or be too subtle to detect, mainly due to floor effects; i.e., increasing the strength of suppression has diminishing returns as neurons approach zero spikes/s, whereas stimulation has a much higher ceiling. Also working against us here is the fact that current viral vector technology limits the percentage of neurons that express the transgene. Still, we hope to address these issues with future experiments that include more systematic variations in light power, and better reagents.

The histology on the Jaws expression is an important aspect of the results. An obvious concern is that the histology was done in area LIP not area MT, and it is not obvious to me whether the expression patterns in area MT would be expected to closely follow those in LIP or whether they might end up being very different. Are there neuronal cell type similarities (or differences) between MT and LIP that should be pointed out when interpreting these results?

We agree with this reviewer that presenting some histology will be useful to readers. We also agree with all three reviewers that there are limits to what one can infer from expression in a different brain area of a different monkey. The revised manuscript now clarifies these limitations. We are not aware of any studies that investigate the similarities or differences in cell types or laminar organization of MT and LIP, but if there is such a study we would gladly cite it.

Unless I missed it, the paper presents the effects of photosuppression only for the trials that included the opt-out option. What effects do you find if the opt-out is not available? Does this substantially change the size of the effect?

We apologize for leaving this ambiguous. In fact, to maximize statistical power, we pooled together trials with and without the sure-bet option for all graphs and analyses of choice data, with the exception of Figure 1C. We have now clarified this in the Materials and methods (subsection “Data analysis”, first paragraph). If we analyze these conditions separately, it turns out that the photosuppression effects were about twice as large on trials where the sure-bet option was *un*available compared to when it was available. We don’t know how seriously to take this result since the effect sizes were modest overall. One possible interpretation is that the sure-bet option absorbed some of the potential biasing effect – that is, some choices that would have been biased away from the preferred direction were instead rendered as opt-outs. But this would seem to predict an overall increase in opt-out choices on laser trials, which we did not observe (if anything, it was the opposite). We have added this point to the newly rewritten Discussion section regarding possible deviations from a straightforward model in which confidence and choice are yoked via a common decision variable (subsection “A common mechanism for choice and confidence”).